



# Sources and formation of carbonaceous aerosols in Xi'an, China: primary emissions and secondary formation constrained by radiocarbon

Haiyan Ni[1,2,3], Ru-Jin Huang[1*], Junji Cao[1], Jie Guo[1], Haoyue Deng[2], Ulrike Dusek[2]

[1]State Key Laboratory of Loess and Quaternary Geology, Key Laboratory of Aerosol Chemistry and Physics, Center for Excellence in Quaternary Science and Global Change, Institute of Earth Environment, Chinese Academy of Sciences, Xi'an, 710061, China
[2]Centre for Isotope Research (CIO), Energy and Sustainability Research Institute Groningen (ESRIG), University of Groningen, Groningen, 9747 AG, the Netherlands
[3]University of Chinese Academy of Sciences, Beijing, 100049, China

*Correspondence to*: rujin.huang@ieecas.cn

**Abstract.** To investigate the sources and formation mechanisms of carbonaceous aerosols, a major contributor to severe particulate air pollution, radiocarbon ([14]C) measurements were conducted on aerosols sampled from November 2015 to November 2016 in Xi'an, China. Based on the [14]C content in elemental carbon (EC), organic carbon (OC) and water-

insoluble OC (WIOC), contributions of major sources to carbonaceous aerosols are estimated over a whole seasonal cycle: primary and secondary fossil sources, primary biomass burning, and other non-fossil carbon formed mainly from secondary processes. Primary fossil sources of EC were further sub-divided into coal and liquid fossil fuel combustion by complementing [14]C data with stable carbon isotopic signatures.

The dominant EC source was liquid fossil fuel combustion (i.e., vehicle emissions), accounting for 64% (median; 45–74%,

interquartile range) of EC in autumn, 60% (41–72%) in summer, 53% (33–69%) in spring and 46% (29–59%) in winter, respectively. An increased contribution from biomass burning to EC was observed in winter (~28%) compared to other seasons (warm period; ~15%). In winter, coal combustion (~25%) and biomass burning equally contributed to EC, whereas in the warm period, coal combustion accounted for a larger fraction of EC than biomass burning. The relative contribution of fossil sources to OC was consistently lower than that to EC, with an annual average of 47 ± 4%. Non-fossil OC of secondary

origin was an important contributor to total OC (35 ± 4%) and accounted for more than half of non-fossil OC (67 ± 6%) throughout the year. Secondary fossil OC (SOC$_{fossil}$) concentrations were higher than primary fossil OC (POC$_{fossil}$) concentrations in winter, but lower than POC$_{fossil}$ in the warm period.

Fossil WIOC and water-souble OC (WSOC) have been widely used as proxies for POC$_{fossil}$ and SOC$_{fossil}$, respectively. This assumption was evaluated by (1) comparing their mass concentrations with POC$_{fossil}$ and SOC$_{fossil}$, and (2) comparing ratios

of fossil WIOC to fossil EC to typical primary OC to EC ratios from fossil sources including both coal combustion and vehicle emissions. The results suggest that fossil WIOC and fossil WSOC are probably a better approximation for primary and secondary fossil OC, respectively, than POC$_{fossil}$ and SOC$_{fossil}$ estimated using the EC tracer method.



## 1. Introduction

Carbonaceous aerosols are an important component of $PM_{2.5}$ (particles with aerodynamic diameter <2.5 µm), constituting typically 20–50% of $PM_{2.5}$ mass in many urban areas in China (Cao et al., 2012; R. J. Huang et al., 2014; Tao et al., 2017). The total carbon content of carbonaceous aerosols (TC) is operationally classified into elemental carbon (EC) and organic carbon (OC) (Pöschl, 2005). EC is emitted as primary aerosols from incomplete combustion of biomass (e.g., wood, crop residues, and grass) and fossil fuels (e.g., coal, gasoline and diesel). In addition to these combustion sources, OC has other non-combustion sources, for example, biogenic emissions, cooking, etc. Unlike EC that is exclusively emitted as primary aerosols, OC includes both primary and secondary OC (POC and SOC, respectively), where SOC is formed in the atmosphere by chemical reaction and gas-to-particle conversion of volatile organic compounds (VOCs) from non-fossil (e.g., biomass burning, biogenic emissions, and cooking) and fossil sources (Jacobson et al., 2000; Kanakidou et al., 2005; Hallquist et al., 2009). Moreover, OC can be separated into water-soluble OC (WSOC) and water-insoluble OC (WIOC), aaccording to water solubility of OC.

High concentrations of carbonaceous aerosols have been observed during severe air pollution events in China (R. J. Huang et al., 2014; Elser et al., 2016; Liu et al., 2016a, 2016b). Knowledge and understanding of the sources and formation processes of carbonaceous aerosols, which remain unclear due to the complicated chemical composition, are highly needed to improve air quality. Clear-cut separation between fossil and non-fossil sources of carbonaceous aerosols can be successfully achieved by radiocarbon measurement (Gustafsson et al., 2009; Szidat et al., 2009; Dusek et al., 2013). Radiocarbon ($^{14}$C) source apportionment exploits the fact that carbonaceous aerosol emitted from fossil sources (e.g., coal combustion, vehicle emissions) does not contain $^{14}$C, whereas carbonaceous aerosol released from non-fossil (or "contemporary") sources has a typical contemporary $^{14}$C signature. Radiocarbon studies show that a sizeable fraction of carbonaceous aerosols is from non-fossil origins, even for aerosols collected in urban areas (Heal, 2014; Cao et al., 2017). For example, Zhang et al. (2015b) found that 48 ± 9% total carbonaceous aerosols were contributed by non-fossil sources in urban areas of 4 large Chinese cities in winter of 2013. $^{14}$C measurements conducted in early winter in 10 Chinese cities show that on average 65 ± 7% total carbonaceous aerosols were derived from non-fossil sources (Liu et al., 2017). When $^{14}$C analysis is conducted for OC and EC separately, contributions from biomass burning and other non-fossil sources to carbonaceous aerosols can be separated for a more comprehensive source apportionment.

$^{14}$C measurements on either WIOC or WSOC can help to separate primary from secondary OC from fossil sources. Fossil sources tend to mainly produce WIOC in primary emissions (Weber et al., 2007; Dai et al., 2015; Yan et al., 2017). Therefore, fossil WIOC ($WIOC_{fossil}$) can be used as a proxy of fossil POC ($POC_{fossil}$). WSOC can be directly emitted as primary aerosols mainly from biomass burning or produced as SOC. There is evidence that SOC produced through the oxidation of VOCs followed by gas-to-particle conversion contains more polar compounds and thus may be an important source of WSOC (Miyazaki et al., 2006; Sannigrahi et al., 2006; Kondo et al., 2007; Weber et al., 2007). Fossil WSOC





(WSOC$_{fossil}$) therefore is thought to be a good proxy of fossil SOC (SOC$_{fossil}$). [14]C analysis of WIOC and WSOC can therefore provide new insights into sources and formation processes of primary and secondary OC, respectively, and has been applied in several source apportionment studies (e.g., Liu et al., 2016a, 2016b; Dusek et al., 2017; Liu et al., 2017). For example, using this approach, Y. L. Zhang et al. (2014) found that secondary fossil OC dominates total fossil OC in a

background site in southern China. Measurements in 4 Chinese megacities highlight the importance of secondary formation to both fossil and non-fossil WSOC in severe winter haze episodes, by combining [14]C measurements of WSOC with positive matrix factorization of aerosol mass spectrometer data (Zhang et al., 2018).

[14]C measurements on EC allow direct separation of fossil and biomass burning source contributions. In addition, analysis of the stable carbon isotopic composition (namely the [13]C/[12]C ratio, expressed as $\delta^{13}C$ in Eq. 1) of EC can be used to separate

fossil sources into coal and liquid fossil fuel combustion (i.e., vehicle emissions), because EC from coal combustion is on average more enriched in the stable carbon isotope [13]C compared to liquid fossil fuel combustion (Andersson et al., 2015; Winiger et al., 2015, 2016; Fang et al., 2018). The interpretation of the stable carbon isotope signature for OC source apportionment is more difficult, because OC is chemically reactive and $\delta^{13}C$ signatures of OC are not only determined by the source signatures but also influenced by chemical reactions of the organic compounds in the atmosphere (Irei et al., 2011;

Pavuluri and Kawamura, 2016).

In this study, one-year PM$_{2.5}$ samples collected from Xi'an, China are investigated. Xi'an is the largest city in northwest China and is also one of the most polluted cities in the world. We present, to our best knowledge, the first 1-year [14]C measurements that distinguish fossil and non-fossil contributions to various carbon fractions, including EC, OC, WIOC and WSOC in Xi'an. Fossil sources of EC are further divided into coal and liquid fossil fuel combustion by complementing

radiocarbon with the stable carbon isotopic signature. Concentrations of POC$_{fossil}$ and SOC$_{fossil}$ are modeled based on the [14]C-apportioned OC and EC and compared with their widely used proxies, i.e., [14]C-apportioned WIOC$_{fossil}$ and WSOC$_{fossil}$, respectively.

## 2. Methods

### 2.1 Sampling

Sampling was conducted in Xi'an, China from 30 November 2015 to 17 November 2016. PM$_{2.5}$ samples were collected on the rooftop (~10 m) of a two-floor building located at the Institute of Earth Environment, Chinese Academy of Sciences (34.2° N, 108.9° E). This site is a typical urban background site surrounded by residential and education areas. The 24 h integrated PM$_{2.5}$ samples were collected from 10:00 to 10:00 the next day (local standard time, LST). PM$_{2.5}$ samples were collected on pre-baked (780 °C for 3 h) quartz fiber filter (QM/A, Whatman Inc., Clifton, NJ, USA, 20.3 cm × 25.4 cm)

using a high-volume sampler (TE-6070 MFC, Tisch Inc., Cleveland, OH, USA) at a flow rate of 1.0 m$^3$ min$^{-1}$. After



collection, the filter sample was immediately removed from the sampler, packed in a pre-baked aluminum foils (450 °C for 3 h), sealed in polyethylene bags and stored in a freezer at -18 °C until analysis.

## 2.2 Thermal/optical organic carbon (OC) and elemental carbon (EC) analysis

Filter pieces of 1.5 cm$^2$ were taken for OC and EC analysis using a carbon analyzer (Model 5L, Sunset Laboratory, Inc., Portland, OR, USA) following the thermal-optical transmittance protocol EUSAAR_2 (Cavalli et al., 2010). In the EUSAAR_2 protocol the filter sample is heated stepwise in an inert helium (He) atmosphere up to 650 °C to thermally desorb organic compounds. After a rapid cooling to 500 °C the sample is heated again stepwise up to 850 °C in an oxidizing 98% He/2% O$_2$ atmosphere to oxidize EC. All carbon gases are converted to CO$_2$ and detected with a non-dispersive infrared (NDIR) detector. During heating in the inert He atmosphere, a fraction of OC pyrolyzes (chars) to light-absorbing EC, as demonstrated by decreasing transmission signal. When the charred OC and original EC are released in the He/O$_2$ atmosphere, transmission signal increases again. The split between OC and EC is set when the transmission signal reaches their pre-pyrolysis value. The sum of OC and EC is total carbon (TC).

At the beginning of each work day, the instrument is calibrated using a sucrose standard solution. The instrument blank, representing the background contamination of the instrument during the analysis, is measured every day and negligible (TC < 0.2 μg m$^{-2}$) compared to the TC loading of the samples (13–246 μg m$^{-2}$; range). The reproducibility determined by duplicate analysis of the filter samples was within 6% for OC and 5% for EC. Details of the OC/EC measurement can also be found in Zenker et al. (2017).

## 2.3 Stable carbon isotopic composition of EC

The stable carbon isotopic composition of EC was measured at the Stable Isotope Laboratory at the Institute of Earth Environment, Chinese Academy of Sciences. To remove OC, filter pieces were heated at 375 °C for 3 h in a vacuum-sealed quartz tube in the presence of CuO catalyst grains. Extraction of EC was done by heating the carbon that remained on the filters at 850 °C for 5 h. The resulting CO$_2$ from EC was isolated by a series of cold traps and quantified manometrically. The stable carbon isotopic composition of the purified CO$_2$ was determined as $\delta^{13}$C ($\delta^{13}$C$_{EC}$ for EC) by offline analysis with a Finnigan MAT-251 mass spectrometer (Bremen, Germany). $\delta^{13}$C values are expressed in the delta notation as per mil (‰) deviation from the international standard Vienna Pee Dee Belemnite (V-PDB):

$$\delta^{13}C\ (‰) = \left[\frac{(^{13}C/^{12}C)_{sample}}{(^{13}C/^{12}C)_{V-PDB}} - 1\right] \times 1000. \qquad (1)$$

A routine laboratory working standard with a known $\delta^{13}$C value was measured every day. The analytical precision of $\delta^{13}$C was better than ± 0.3 ‰ based on duplicate analyses. Details of stable carbon isotope measurements are described in our previous studies (Cao et al., 2011, 2013; Ni et al., 2018).



### 2.4 Radiocarbon ($^{14}$C) measurements of OC, WIOC and EC

#### 2.4.1 Sample selection for $^{14}$C analysis

For $^{14}$C analysis of OC, WIOC and EC, 3 composite samples per season were selected to represent high (H), medium (M) and low (L) concentrations of total carbon (TC = OC + EC), to cover various pollution conditions in each season. Each

composite sample consists of 2 to 4 24 h filter pieces with similar TC loadings and air mass backward trajectories (Fig. S1, Table S1). In total, 36 radiocarbon data were measured, including 12 OC, 12 WIOC and 12 EC. WSOC can be calculated as the difference between OC and WIOC.

#### 2.4.2 Extraction of OC, WIOC and EC

OC, WIOC and EC extractions were conducted on our custom-built aerosol combustion system (ACS). The ACS has been

described in detail by Dusek et al. (2014) and evaluated in two intercomparison studies (Szidat et al., 2013; Zenker et al., 2017). In brief, the ACS consists of a reaction tube and a $CO_2$ purification line. In the reaction tube aerosol filter samples are inserted into a filter holder and heated at different temperatures in pure $O_2$. Combustion products are fully oxidized using a platinum catalyst. The resulting $CO_2$ is separated from other gases (e.g., $NO_x$, water vapor) in the purification line. Here, $NO_x$ and liberated halogens are first removed by a heated oven (650 °C) filled with copper grains and silver, water is then

removed by a U-type tube cooled with dry ice-ethanol mixture (around -70 °C) and a flask containing phosphorus pentoxide ($P_2O_5$). The amount of purified $CO_2$ is determined manometrically in a calibrated volume and $CO_2$ is subsequently stored in flame-sealed glass ampules.

OC is combusted by heating filter pieces at 375 °C for 10 min. WIOC and EC are combusted from water-extracted filter pieces. By water-extraction, water-soluble OC (WSOC) is removed from filter pieces (Dusek et al., 2014). For WIOC, a

water-extracted filter piece is heated at 375 °C for 10 minutes. Subsequently, the oven temperature is increased to 450 °C for 3 min to remove the most refractory OC that left on the filter. However, during this step some less refractory EC might be lost. After this step, OC has been completely removed from the filter pieces. Finally, the remaining EC is combusted by heating the filter at 650 °C in $O_2$ for 5 min (Dusek et al., 2017; Zenker et al., 2017).

Contamination during the extraction procedure is determined by following the same extraction procedures with either empty

filter boat or pre-heated filters (at 650 °C in $O_2$ for 10 min). The contamination yields on average 0.85 µgC OC, 0.73 µgC WIOC and 0.72 µgC EC per extraction, respectively. Compared with our sample size of 45–210 µgC OC, 45–328 µgC WIOC and 15–184 µgC EC, the contamination is relatively small (<5 % of the sample amount).

#### 2.4.3 $^{14}$C measurements by accelerator mass spectrometer (AMS)

$^{14}$C measurements were conducted using the the Mini Carbon Dating System (MICADAS) AMS at the Centre for Isotope

Research at the University of Groningen. The extracted $CO_2$ is released from the glass ampules and captured by a zeolite trap



within a gas inlet system (Ruff et al., 2007), where the sample is diluted using He to 5% $CO_2$ (Salazar et al., 2015). The $CO_2$/He mixture is directly introduced into the Cs sputter ion sources of the MICADAS at a constant rate (Synal et al., 2007).

The $^{14}C/^{12}C$ ratio of an aerosol sample is usually normalized to the $^{14}C/^{12}C$ ratio of an oxalic acid standard (OXII) and expressed as fraction modern ($F^{14}C$). The $^{14}C/^{12}C$ ratio of OXII is related to the unperturbed atmosphere in the reference year

of 1950 by multiplying it with a factor of 0.7459 (Mook and Van Der Plicht, 1999; Reimer et al., 2004):

$$F^{14}C = \frac{(^{14}C/^{12}C)_{sample,[-25]}}{0.7459 \times (^{14}C/^{12}C)_{OXII,[-25]}} \tag{2}$$

where the $^{14}C/^{12}C$ ratio of the sample and OXII are both corrected for machine background and normalized to $\delta^{13}C = -25$ ‰ to correct for isotope fractionation.

The $F^{14}C$ values are corrected for memory effect (Wacker et al., 2010) using alternate measurements of OXII and $^{14}C$-free

material as gaseous standards. Correction for instrument background (Salazar et al., 2015) is done by subtracting the memory corrected $F^{14}C$ values of the $^{14}C$-free standard. Finally, the values are normalized to the average value of the (memory and background corrected) OXII standards. A set of secondary standards is used to assess the robustness and reliability of the data. This includes IAEA-C7 with a consensus value of $F^{14}C = 0.4953 \pm 0.0012$ and sample masses of 76 μg and 80 μg and IAEA-C8 with a consensus value of $F^{14}C = 0.1503 \pm 0.0017$ and sample masses of 63 μg and 100 μg. All standards

including OXII and $^{14}C$-free material used for data correction and IAEA-C7 and IAEA-C8 for quality control of AMS measurements are measured on the same day as the samples. $F^{14}C$ values of secondary standards undergo the same data correction as the samples. Results of IAEA-C7 and C8 agree within uncertainties (Table S2).

$F^{14}C$ of carbon from fossil sources is 0, and carbon from non-fossil sources (or "contemporary" sources) should have $F^{14}C$ of 1. But the extensive release of $^{14}C$ from nuclear bomb tests in the late 1950s and early 1960s and $^{14}C$-free $CO_2$ from fossil

fuel combustion has perturbed the atmospheric $F^{14}C$ values significantly. The former increased the $F^{14}C$ in the atmosphere by up to a factor of 2 in the northern hemisphere in the 1960s. The nuclear tests have been banned in the atmosphere, outer space and under water since 1963. Since then, the atmospheric $F^{14}C$ has been slowly decreasing, as $^{14}C$ is mainly taken up by the oceans and terrestrial biosphere and diluted by $^{14}C$-free $CO_2$ (Hua and Barbetti, 2004; Levin et al., 2010). Currently, the $F^{14}C$ of the atmospheric $CO_2$ is approximately 1.04 (Levin et al., 2008).

**2.5 Estimation of source contributions to different carbon fractions**

$F^{14}C$ of EC, OC and WIOC (i.e., $F^{14}C_{(EC)}$, $F^{14}C_{(OC)}$ and $F^{14}C_{(WIOC)}$, respectively) are directly measured. Mass concentrations ($M_{WSOC}$) and $F^{14}C$ of WSOC ($F^{14}C_{(WSOC)}$) can be calculated as

$$M_{WSOC} = M_{OC} - M_{WIOC} \tag{3}$$



$$F^{14}C_{(WSOC)} = \frac{F^{14}C_{(OC)} \times M_{OC} - F^{14}C_{(WIOC)} \times M_{WIOC}}{M_{OC} - M_{WIOC}}. \qquad (4)$$

where $M_{OC}$ and $M_{WIOC}$ are mass concentrations of OC and WIOC, respectively. $M_{OC}$ is measured by the thermal-optical method as described in Sect. 2.2.

To estimate $M_{WIOC}$, we assume two extreme cases following the method of Dusek et al. (2017). (1) WIOC is completely recovered. That is, the recovery of WIOC is 100%, where the recovery is estimated by dividing the WIOC mass extracted using ACS ($M_{WIOC,e}$) with the WIOC mass in the aerosol samples. But the WIOC combustion temperature of 375 °C in the ACS is highly likely not high enough to recover 100 % of WIOC. Thus, this estimation is an underestimate of $M_{WIOC}$ ($M1_{WIOC}$). (2) We assume that WIOC has the same recovery as OC. The $M_{WIOC}$ can be calculated by dividing $M_{WIOC,e}$ by the OC recovery. Due to the fact that usually less WIOC than OC is lost to charring, this probably is an overestimate of $M_{WIOC}$ ($M2_{WIOC}$). $M_{WIOC}$ is assumed to vary from $M1_{WIOC}$ to $M2_{WIOC}$. The most likely value of $M_{WIOC}$ is chosen at $M1_{WIOC}+2/3\times(M2_{WIOC}-M1_{WIOC})$, because it is more likely that WIOC has a similar recovery as OC rather than 100 % recovery. Once $M_{WIOC}$ is estimated, the $F^{14}C_{(WSOC)}$ can be calculated following the Eq. (4). The best estimate and ranges of $F^{14}C_{(WSOC)}$ is presented in Fig. S2 and Table S1.

$F^{14}C_{(EC)}$ can be converted to the relative contribution of biomass burning to EC ($f_{bb}(EC)$) by dividing with $F^{14}C$ of biomass burning ($F^{14}C_{bb} = 1.10 \pm 0.05$; (Lewis et al., 2004; Mohn et al., 2008; Palstra and Meijer, 2014), to eliminate the effect from nuclear bomb tests in the 1960s. Analogously, the relative contribution of non-fossil sources to OC, WIOC and WSOC (i.e., $f_{nf}(OC)$, $f_{nf}(WIOC)$ and $f_{nf}(WSOC)$, respectively) can be estimated from their corresponding $F^{14}C$ values and $F^{14}C$ of non-fossil sources ($F^{14}C_{nf}=1.09 \pm 0.05$; Lewis et al., 2004; Levin et al., 2010; Y. L. Zhang et al., 2014;). The lower limit of $F^{14}C_{nf}$ corresponds to current biospheric sources as the source of OC (1.04), and the upper limit corresponds to wood combustion as the main source of OC, with only minor contribution from annual crops.

EC is primarily produced from biomass burning ($EC_{bb}$) and fossil fuel combustion ($EC_{fossil}$), and absolute EC concentrations from each source can be estimated as:

$$EC_{bb} = M_{EC} \times f_{bb}(EC) \qquad (5)$$

$$EC_{fossil} = M_{EC} \times \left(1 - f_{bb}(EC)\right) = M_{EC} \times f_{fossil}(EC) \qquad (6)$$

where $f_{fossil}(EC)$ is the relative contribution of fossil sources to EC, $M_{EC}$ are mass concentrations of EC. Analogously, mass concentrations of OC, WIOC and WSOC from non-fossil sources ($OC_{nf}$, $WIOC_{nf}$ and $WSOC_{nf}$, respectively) and fossil sources ($OC_{fossil}$, $WIOC_{fossil}$ and $WSOC_{fossil}$, respectively) can be determined.

More detailed source apportionment of OC can be achieved by combining $^{14}C$-apportioned OC and EC with characteristic primary OC/EC ratios for each source (i.e., using EC as a tracer of primary emissions; EC tracer method). Biomass burning





usually has higher primary OC/EC ratios ($r_{bb}$ = 3–10) than those for coal combustion ($r_{coal}$ = 1.6–3) and vehicle exhausts ($r_{vehicle}$ = 0.5–1.3) (Ni et al. (2017) and references therein). Best estimate of $r_{bb}$ (4 ± 1; average ± SD), $r_{coal}$ (2.38 ± 0.44), and $r_{vehicle}$ (0.85 ± 0.16) is done through a literature search as described in Ni et al. (2018) and comparable to values used in ealier [14]C source apportionment in China (Y. L. Zhang et al., 2014, 2015a).

Primary biomass burning OC ($POC_{bb}$) can be estimated by multiplying $EC_{bb}$ with $r_{bb}$:

$$POC_{bb} = EC_{bb} \times r_{bb} \qquad (7)$$

Other non-fossil OC excluding $POC_{bb}$ ($OC_{o,nf}$) can be estimated as:

$$OC_{o,nf} = OC_{nf} - POC_{bb} \qquad (8)$$

where $OC_{o,nf}$ includes OC from all non-fossil sources other than primary biomass burning, thus mainly consists of secondary
OC from biomass burning ($SOC_{bb}$), primary and secondary biogenic OC, as well as cooking emissions. In most cases, contributions of primary biogenic OC to $PM_{2.5}$ are likely small.

$OC_{fossil}$ includes both primary and secondary OC from fossil sources ($POC_{fossil}$ and $SOC_{fossil}$, respectively):

$$OC_{fossil} = POC_{fossil} + SOC_{fossil}, \qquad (9)$$

where $POC_{fossil}$ can be estimated from $EC_{fossil}$ and primary OC/EC ratio of fossil fuel combustion ($r_{fossil}$):

$$POC_{fossil} = EC_{fossil} \times r_{fossil}. \qquad (10)$$

Fossil sources in China are almost exclusively from coal combustion and vehicle emissions, thus $r_{fossil}$ can be estimated as

$$r_{fossil} = r_{coal} \times p + r_{vehicle} \times (1 - p), \qquad (11)$$

where $p$ is the relative contribution of coal combustion to fossil EC. That is, $p$ = $EC_{coal}/EC_{fossil}$, where estimation of $EC_{coal}$ is achieved by combing $F^{14}C_{(EC)}$ and $\delta^{13}C_{EC}$ with the Bayesian calculations as described in details in the Sect. 2.6 and
Supplement S1.

To propagate uncertainties, a Monte Carlo simulation with 10000 individual calculations was conducted. For each individual calculation, $F^{14}C_{(EC)}$, $F^{14}C_{(OC)}$, $F^{14}C_{(WIOC)}$ and concentrations of EC, OC and WIOC are randomly chosen from a normal distribution symmetric around the measured values with the experimental uncertainties as standard deviation (SD). For $F^{14}C_{bb}$, $F^{14}C_{nf}$, $r_{bb}$, $r_{coal}$ and $r_{vehicle}$ random values are chosen from a triangular frequency distribution with its maximum at the
central value and is 0 at the lower limit and upper limit. For $p$ values, random values from the respective PDF of $p$ were used (Supplement S1). In this way 10000 random sets of variables can be generated. For $f_{bb}$(EC), $f_{nf}$(OC), $f_{nf}$(WIOC), $f_{nf}$(WSOC), $EC_{bb}$, $EC_{fossil}$, $OC_{nf}$, $OC_{fossil}$, $WIOC_{nf}$, $WIOC_{fossil}$, $WSOC_{nf}$, $WSOC_{fossil}$, $POC_{bb}$ and $OC_{o,nf}$, the derived average represents the best estimate, and the SD represents the combined uncertainties (Tables S3, S4). For $POC_{fossil}$ and $SOC_{fossil}$, the median value





is considered as the best estimates and the interquartile range (25th–75th percentile) are used as uncertainties, because the PDFs of $POC_{fossil}$ and $SOC_{fossil}$ are asymmetric (Fig. S3b, Table S5).

**2.6 Source apportionment of EC using Bayesian statistics**

Using $F^{14}C$ and $\delta^{13}C$ signatures of EC ($F^{14}C_{(EC)}$, $\delta^{13}C_{EC}$) and assuming isotope mass balance in combination with a Bayesian

Markov chain Monte Carlo (MCMC) scheme, it is possible to differentiate the 3 main sources of EC: biomass burning, liquid fossil fuel combustion (i.e., vehicle emissions) and coal combustion (Andersson et al., 2015; Li et al., 2016; Winiger et al., 2016; Fang et al., 2018):

$$\begin{pmatrix} F^{14}C_{(EC)} \\ \delta^{13}C_{EC} \\ 1 \end{pmatrix} = \begin{pmatrix} F^{14}C_{bb} & F^{14}C_{liq.fossil} & F^{14}C_{coal} \\ \delta^{13}C_{bb} & \delta^{13}C_{liq.fossil} & \delta^{13}C_{coal} \\ 1 & 1 & 1 \end{pmatrix} \begin{pmatrix} f_{bb} \\ f_{liq.fossil} \\ f_{coal} \end{pmatrix} \qquad (12)$$

where the last row ensures the mass balance; $f_{bb}$, $f_{liq.fossil}$ and $f_{coal}$ are the relative contribution from biomass burning, liquid

fossil fuel combustion and coal combustion to EC, respectively; $F^{14}C_{bb}$ is the $F^{14}C$ of biomass burning ($1.10 \pm 0.05$), as mentioned in Sect. 2.5. $F^{14}C_{liq.fossil}$ and $F^{14}C_{coal}$ are zero due to the long-time decay. $\delta^{13}C_{bb}$, $\delta^{13}C_{liq.fossil}$ and $\delta^{13}C_{coal}$ are the $\delta^{13}C$ signature of EC emitted from biomass burning, liquid fossil fuel combustion and coal combustion, respectively. The means and the standard deviations for $\delta^{13}C_{bb}$ ($-26.7 \pm 1.8$ ‰ for C3 plants, and $-16.4 \pm 1.4$ ‰ for corn stalk), $\delta^{13}C_{liq.fossil}$ ($-25.5 \pm 1.3$ ‰), and $\delta^{13}C_{coal}$ ($-23.4 \pm 1.3$ ‰) are compiled and established by literature studies in previous publications (Andersson et

al. (2015) and references therein; Ni et al., 2018). The MCMC technique takes into account the variability in the source signatures of $F^{14}C$ and $\delta^{13}C$ (Parnell et al., 2010, 2013), where $\delta^{13}C$ introduces a larger uncertainty than $F^{14}C$ as $\delta^{13}C$ varies with fuel types and combustion conditions. The results of the MCMC calculations are the posterior probability density functions (PDFs) for $f_{bb}$, $f_{liq.fossil}$ and $f_{coal}$ (Fig. S4). The median was used to represent the best estimate of the $f_{bb}$, $f_{liq.fossil}$ and $f_{coal}$. Uncertainties of this best estimate are expressed as an interquartile range (25th-75th percentile) of the corresponding

PDFs. The MCMC-derived $f_{bb}$ (calculated by Eq. 12) is very similar to that obtained from radiocarbon data ($f_{bb}(EC)$, Eq. 5) as both of them are well constrained by $F^{14}C$. In this study, $f_{bb}$ and $f_{bb}(EC)$ are therefore used interchangeably. Details on the MCMC-driven Bayesian approach have been described in our earlier study (Ni et al., 2018).

**3 Results**

**3.1 $^{14}C$-based source apportionment of EC and OC**

EC is derived mainly from fossil sources, regardless of differences in EC concentrations and seasonal variations. The relative contribution of fossil fuel combustion to EC ($f_{fossil}(EC)$) ranges from 69% to 89%, with an annual average of $82 \pm 6\%$ (Fig. 1a). The relative contribution of fossil sources to OC ($f_{fossil}(OC)$ is consistently smaller than $f_{fossil}(EC)$ (Fig. 1b). The values of $f_{fossil}(OC)$ range from 41% to 51%, with an annual average of $47 \pm 4\%$. The absolute difference in the fossil fractions



between OC and EC is on average 35% (28%–42%; range). The main reason for this difference is that biomass burning emits more OC relative to EC compared to the fossil sources (Streets et al., 2003; Akagi et al., 2011; Zhou et al., 2017). Thus, even if biomass burning contributes a small fraction to EC, it will have a much higher contribution to primary OC. Additionally other non-fossil sources, such as secondary biomass burning emissions, primary and secondary biogenic

emissions as well as cooking contribute to OC, but not to EC.

The annual average $f_{fossil}(EC)$ and $f_{fossil}(OC)$ reported here is consistent with the results reported at an urban site of the same Chinese city in 2008/2009 ($f_{fossil}(EC) = 83 \pm 5\%$, $f_{fossil}(OC) = 46 \pm 8\%$; Ni et al., 2018), an urban site of Beijing, China in 2013/2014 ($f_{fossil}(EC) = 82 \pm 7\%$, $f_{fossil}(OC) = 48 \pm 12\%$; Zhang et al., 2017) and 2010/2011 ($f_{fossil}(EC) = 79 \pm 6\%$; Zhang et al., 2015b) and a background receptor site of Ningbo, China ($f_{fossil}(EC) = 77 \pm 15\%$; Liu et al., 2013). Much lower $f_{fossil}(EC)$

and $f_{fossil}(OC)$ was found at a regional background site in South China in 2005/2006 ($f_{fossil}(EC) = 38 \pm 11\%$ and $f_{fossil}(OC) = 19 \pm 10\%$ for Hainan; Y. L. Zhang et al., 2014), regional receptor sites in South Asia in 2008/2009 ($f_{fossil}(EC) = 27 \pm 6\%$ and $f_{fossil}(OC) = 31 \pm 5\%$ for Hanimaadhoo, Maldives and $f_{fossil}(EC) = 41 \pm 5\%$ and $f_{fossil}(OC) = 36 \pm 5\%$ for Sinhagad, India; Sheesley et al., 2012), where regional/local biomass burning contributes much more to carbonaceous aerosols than fossil fuel combustion and the [14]C levels can change significantly with the origin of air masses.

The $f_{fossil}(EC)$ and $f_{fossil}(OC)$ follow the same seasonal trends: the values are lower in winter and higher in the rest of the seasons (i.e., warm period). In the warm period, there is a slight but consistent tendency to be higher in spring in general and also to be slightly lower under the cleanest periods (Fig. 1, Tables S3, S6). The low $f_{fossil}(EC)$ in winter is due to the substantially increased contribution from biomass burning (mainly wood burning) for heating in winter, which gradually stops in spring but in summer and early autumn, open biomass burning (mainly crop residues) occurs in Xi'an and its

surrounding areas. Some biomass burning for cooking is probably present all year round (Huang et al., 2012; T. Zhang et al., 2014;). The seasonality in biomass burning activity is consistent with the variations of $f_{bb}(EC)$, which is higher in winter ($28 \pm 4\%$) and lower in other seasons (around 15%) with a slightly lower values in spring ($14 \pm 3\%$). This is in line with our previous study in Xi'an, China in 2008/2009 (Ni et al., 2018). Beijing shows a very different seasonal trend, where $f_{bb}(EC)$ was lowest in summer (~7%) and increased to ~20% during the rest of the year (Zhang et al., 2017). The distinct different

values and seasonality of $f_{bb}(EC)$ in Xi'an and Beijing indicate that biomass burning emissions are seasonally dependent and their influences vary spatially in different Chinese cities. The seasonal trends of $f_{fossil}(OC)$ were different in Beijing as well, with higher $f_{fossil}(OC)$ in winter than in other seasons (Yan et al., 2017; Zhang et al., 2017). This is in line with previous source apportionment results that during wintertime biomass burning is a major source of OC in Xi'an and coal combustion is a dominant source for OC in Beijing (R. J. Huang et al., 2014; Elser et al., 2016).

EC concentrations from fossil fuel combustion (EC$_{fossil}$) span a range from around 0.6 to 7 µg m$^{-3}$ and increase by roughly a factor of 3 from summer to winter when separately comparing clean and polluted periods. The remaining EC is contributed by biomass burning (EC$_{bb}$), which varies in a wider range than EC$_{fossil}$ from around 0.1 to 3 µg m$^{-3}$ (Fig. 1a, Table S4).


$EC_{fossil}$ values are on average 2–3 times higher than $EC_{bb}$ in winter and 5–8 times higher in other seasons. This implies that the winter-summer differences in biomass burning emissions is larger than fossil fuel combustion emissions, regardless of the fact that both biomass burning and coal combustion are expected to increase during wintertime for heating (T. Zhang et al., 2014; Shen et al., 2017; Zhu et al., 2017). OC concentrations from fossil fuel combustion ($OC_{fossil}$) range from around 1

to 20 µg m⁻³, with an annual average of 6.8 ± 6.0 µg m⁻³, which is comparable to non-fossil OC concentrations (range: 2–28 µg m⁻³; mean: 8.2 ± 8.2 µg m⁻³). Clear seasonal variations were observed in both EC and OC from fossil and non-fossil sources, with maxima in winter and minima in summer (Table S7). This is mainly because of an increase in coal burning and biomass burning for heating as well as unfavorable meteorological conditions in winter.

### 3.2 ¹⁴C- based source apportionment of water-soluble and water-insoluble OC

The fossil contribution to total WIOC ($f_{fossil}$(WIOC)) varied from 49 ± 1% in winter to 60 ± 5% in summer, with an annual average of 55 ± 5%. In winter the enhanced biomass burning is a source of non-fossil WIOC (Dusek et al., 2017). The relative contributions of fossil sources to WSOC ($f_{fossil}$(WSOC) = 42 ± 6%) were smaller than that to WIOC for nearly all the samples throughout the year. In winter both primary emission and secondary formation from biomass burning contribute to WSOC and in the warm period additionally biogenic SOC, though the latter concentrations are probably relatively low. In

addition, primary fossil emissions are expected to contribute very little to WSOC, so the lower fossil fractions in WSOC are in line with expectations. In this study, the largest differences between fossil fractions in WIOC and WSOC were found to be 36% for sample Summer-L (e.g., low TC concentrations in summer). Summer-L had the lowest $f_{fossil}$(WSOC) of 28 ± 2% (Fig. 2a), which was contrary to the stable $f_{fossil}$(EC) in the warm period (Fig. 1a) and therefore cannot be explained by an increase in primary (or probably secondary) biomass burning OC. This indicates that the lowest $f_{fossil}$(WSOC) for Summer-L

was probably due to the impact of biogenic OC in the clean period.

As shown in Fig. 2a, WSOC concentrations from non-fossil sources ($WSOC_{nf}$) are larger than WSOC from fossil sources ($WSOC_{fossil}$), with an annual average of 5.1 ± 4.9 µg m⁻³ for $WSOC_{nf}$ versus an average of 3.6 ± 3.0 µg m⁻³ for $WSOC_{fossil}$. WIOC concentrations from non-fossil sources ($WIOC_{nf}$) are comparable with those from fossil sources ($WIOC_{fossil}$). $WSOC_{nf}$, $WSOC_{fossil}$, $WIOC_{nf}$ and $WIOC_{fossil}$ show the same seasonal trends, with higher mass concentrations in winter and lower in

the warm period. $WSOC_{nf}$ is responsible for ~ 35% of the increased OC mass in winter, followed by $WIOC_{nf}$ (~24%), $WIOC_{fossil}$ (~ 22%) and $WSOC_{ff}$ (~ 19%).

Figure 2b shows the fraction of $WIOC_{nf}$, $WSOC_{nf}$, $WIOC_{fossil}$ and $WSOC_{fossil}$ in the total OC in different seasons. WSOC (the sum of the blue areas) on yearly average accounted for 60 ± 5% of OC (ranging from 53–70%), consistent with previous measurements in Xi'an (Cheng et al., 2013; Zhang et al., 2018; Zhao et al., 2018). The remaining 40 ± 5% of OC is WIOC

(the sum of red areas). Throughout the year, $WSOC_{nf}$ was the largest contributor to OC, which accounts for about one-third of the total OC, probably resulting from the mostly water-soluble biomass-burning POC and SOC as well as biogenic SOC





(e.g., Mayol-Bracero et al., 2002; Nozière et al., 2015; Dusek et al., 2017).The respective proportions of $WSOC_{fossil}$, $WIOC_{fossil}$ and $WIOC_{nf}$ in OC were 26 %, 21% and 17% on a yearly average in descending order, very likely related to secondary fossil OC, primary fossil OC and primary biomass burning, respectively (Weber et al., 2007; Dai et al., 2015; Dusek et al., 2017; Yan et al., 2017).

The majority (60–76%) of the non-fossil OC was water-soluble. This result is qualitatively consistent with findings reported for an urban site of Xi'an (Zhang et al., 2018) and other places such as at an urban site of Beijing, China (Zhang et al., 2018), an urban or rural site in Switzerland (Zhang et al., 2013), a remote site on Hainan Island, southern China (Y. L. Zhang et al., 2014) and two rural sites in the eastern United States (Wozniak et al., 2012) and a regional background site in the Netherlands (Dusek et al., 2017). Seasonal variations of $(WSOC/OC)_{nf}$ ratios were also observed, with lower ratios in winter

(around 0.6) and higher ratios in summer and spring (around 0.7). This reflects the higher fraction of $WIOC_{nf}$ in $OC_{nf}$ during wintertime, resulting from primary biomass burning emissions (Dusek et al., 2017). In summer and spring, concentrations of $WSOC_{nf}$ and $OC_{nf}$ are both small and the contribution of biogenic SOC to $WSOC_{nf}$ can be noticeable (Dusek et al., 2017).

The fossil OC is less water soluble in winter with lower $(WSOC/OC)_{fossil}$ ratios of around 0.5 than in the warm period (Fig. 3). $WSOC_{fossil}$ can come mainly from secondary formation and/or photochemical aging of primary organic aerosols, thus the

higher $(WSOC/OC)_{fossil}$ ratios in the warm period suggest an enhanced SOC formation from fossil VOCs from vehicle emissions and/or coal burning. In spring and summer there is a clear increasing tend of $(WSOC/OC)_{fossil}$ in more polluted periods. Elevated $(WSOC/OC)_{fossil}$ ratios in polluted periods are very likely related to the formation of high pollutant concentrations in spring and summer. More stagnant conditions in the polluted periods (indicated by lower wind speed, see Fig. 3) that allow for accumulation of pollutants also provide more time for photochemical processes and SOC formation. As

a consequence, formation of fossil WSOC will increase in stagnant conditions. At the same time, $(WIOC/EC)_{fossil}$ ratios decline when pollution gets worse, suggesting removal of WIOC, likely through photochemical reactions. This can shift the water-soluble vs. water-insoluble distribution for fossil OC to WSOC (Szidat et al., 2009). As a consequence, the $(WSOC/OC)_{fossil}$ ratio is higher for Summer-H (70%) than for Summer-L (52%).

### 3.3 Combustion sources apportioned by stable carbon isotopes

Along with radiocarbon data, the stable carbon isotopic ratio of EC (denoted by $\delta^{13}C_{EC}$) provides additional insight into source apportionment of EC, especially between different type of fossil sources (i.e., coal versus liquid fossil fuel combustion). Figure 4 shows $^{14}C$-based $f_{fossil}(EC)$ against $\delta^{13}C_{EC}$ in Xi'an in different seasons for 2015/2016 from this study and in winter for 2008/2009 from Ni et al. (2018), together with the ranges of endmembers (i.e., isotopic signature) for the different EC sources of coal combustion, liquid fossil fuel combustion and biomass burning (C3 and C4 plants). $f_{fossil}(EC)$ is

well constrained, clearly separating fossil sources from biomass burning. In contrast to $^{14}C$, the source endmembers (i.e., isotopic signature) for $\delta^{13}C$ are less well constrained and $\delta^{13}C$ values for liquid fossil fuel combustion overlap with $\delta^{13}C$



values for both coal and C3 plant combustion. Regardless of the changes of $\delta^{13}C_{EC}$ in different seasons, all the $\delta^{13}C_{EC}$ data points fall within the range of burning C3 plant, coal and liquid fossil fuel, indicating that C3 plant is the dominating biomass type in Xi'an with little influence from C4 plant burning. In Xi'an, the dominant C4 plant is corn stalk, which is burned for cooking and heating in the areas surrounding Xi'an (Sun et al., 2017; Zhu et al., 2017).

The annually averaged $\delta^{13}C_{EC}$ is -24.9 ± 0.4 ‰ (± SD). Moderate seasonal variation of $\delta^{13}C_{EC}$ was observed, reflecting a moderate shift in the relative contributions from combustion sources throughout the year. The $\delta^{13}C_{EC}$ in autumn (-25.3 ± 0.2 ‰) and summer (-25.0 ± 0.3 ‰) are most depleted and fall into the overlapped $\delta^{13}C$ range for liquid fossil fuel combustion and C3 plant burning. Because the $^{14}C$ values in autumn and summer indicate that biomass burning contribution to EC is relatively low (~16%), we can expect that liquid fossil fuel combustion dominates EC in autumn and summer.

$\delta^{13}C_{EC}$ signatures in winter (-24.8 ± 0.2 ‰) scatter into the range for C3 plant, liquid fossil fuel and coal combustion, implying that EC is influenced by mixed sources. The $\delta^{13}C_{EC}$ signatures in spring (-24.6 ± 0.3 ‰) overlaps with both liquid fossil fuel combustion and coal combustion. Only the sample Spring-L is characterized by the most enriched $\delta^{13}C_{EC}$ value among all the samples, even more enriched than wintertime $\delta^{13}C_{EC}$, when coal combustion for heating is expected to influence EC strongly. At the same time, higher contributions from biomass burning (i.e., lower $f_{fossil}$(EC) ) were observed

for Spring-L. This suggests contributions from a $^{13}C$-enriched biomass burning, that is, corn stalk burning (C4 plant). The contribution of this regional source can become noticeable in the relatively clean air that characterizes Spring-L.

To estimate seasonal source contributions to EC, we combined all the data points from each season for the Bayesian Markov chain Monte Carlo techniques (MCMC) calculations. The MCMC results (Fig. 5a, Fig. S4, Table S8) show that the dominant EC source is liquid fossil fuel combustion (i.e., vehicle emissions). Liquid fossil fuel combustion accounts for 64 % (median;

45–74%, interquartile range) of EC in autumn, 60% (41–72%) in summer, 53% (33–69%) in spring, and 46% (29–59%) in winter, respectively, in descending order. Biomass burning EC is a small fraction of total EC throughout the year. However, the relative contribution of biomass burning to EC increased in winter (28 %; 26–31%), and is comparable to the relative contribution of coal combustion (25%; 13–41%). In the warm period, coal combustion for cooking accounts for a larger fraction of EC than biomass burning.

EC concentrations from biomass burning ($EC_{bb}$) increased by 9 times from summer (seasonal average of 0.2 μg m$^{-3}$) to winter (1.8 μg m$^{-3}$; Fig. 5b, Table S9). EC from coal combustion ($EC_{coal}$) has a 5-fold increase from around 0.3 μg m$^{-3}$ in summer and autumn to 1.6 μg m$^{-3}$ in winter.  EC from liquid fossil fuel ($EC_{liq.fossil}$) varies less strongly than $EC_{bb}$ and $EC_{coal}$, by 4-times from 0.7 μg m$^{-3}$ in summer and 2.9 μg m$^{-3}$ in winter. Liquid fossil fuel combustion (i.e., vehicle emissions) should be roughly constant throughout the year. The increased concentrations of $EC_{liq.fossil}$ in winter are most likely due to

unfavorable meteorological conditions. An increase larger than a factor of 4 therefore suggests increasing emissions in winter. Compared to the increase in $EC_{liq.fossil}$, $EC_{coal}$ only increases moderately in winter, reflecting the moderate seasonal





variation of $\delta^{13}C_{EC}$ (Fig. 4). This suggests that coal combustion is a more constant source over the year 2015/2016. Coal use for heating during wintertime has been decreasing since the year 2008/2009 (Ni et al., 2018), suggested by the more depleted wintertime $\delta^{13}C_{EC}$ in 2015/2016 than that in 2008/2009 (Fig. 4). The decreasing contribution from coal combustion to EC is consistent with the changes in energy consumption and the decreasing concentrations of coal combustion indicators (e.g., As

and Pb) in Xi'an as found in pervious studies (Xu et al., 2016; Ni et al., 2018). The poor separation of fossil sources of EC into coal combustion and liquid fossil fuel combustion could be another reason, but it is difficult to quantify this effect due to our poor knowledge of $\delta^{13}C$ source endmembers.

### 3.4 Primary and secondary OC

Based on the EC tracer method, $OC_{o,nf}$ is representative of $SOC_{nf}$, or can be considered an upper limit of $SOC_{nf}$ if cooking

sources are significant. The fractions of primary OC ($POC_{bb}$ and $POC_{fossil}$) and secondary OC ($OC_{o,nf}$, and $SOC_{fossil}$) in total OC are shown in Figure 6 and Table S5. On a yearly basis, the most important contributor to OC was $OC_{o,nf}$ (around 35%). For all samples, $OC_{o,nf}$ concentrations were higher than $POC_{bb}$, despite the wide range of total OC concentrations in different seasons. $POC_{bb}$ contributed a relatively small fraction of OC (15–18%) in the warm period, which increased to 22% during winter when Xi'an was impacted significantly by biomass burning for heating and cooking. Enhanced biomass burning

activities during wintertime in Xi'an have also been reported by measurements of markers for biomass burning such as levoglucosan and $K^+$ (T. Zhang et al., 2014; Shen et al., 2017). In winter, $SOC_{fossil}$ was generally more abundant than $POC_{fossil}$, suggesting that secondary formation rather than primary emissions was a more important contributor to total $OC_{fossil}$. However, in the warm period, for fossil fuel derived OC ($POC_{fossil}$ and $SOC_{fossil}$), primary emissions dominated over secondary formation (Figs. 6b, 6c). The $SOC_{fossil}/OC_{fossil}$ ratios indicate that $SOC_{fossil}$ contributes roughly 57% to $OC_{fossil}$ in

winter versus 37% in the warm period. However, the lower $SOC_{fossil}/OC_{fossil}$ ratios in the warm period (especially in summer) than winter in this study is unexpected due to the favorable atmospheric conditions (e.g., higher temperature and stronger solar radiation). Much higher contribution of $SOC_{fossil}$ to $OC_{fossil}$ (an annual average of around 70%) was found in southern China (Y. L. Zhang et al., 2014). The importance of fossil derived SOC formation to fossil OC during wintertime was also found in other Chinese cities, including Beijing, Shanghai and Guangzhou (Zhang et al., 2015a), suggesting the rapid

formation of SOC even in winter (R. J. Huang et al., 2014).

As for OC from secondary origin (i.e., $SOC_{fossil}$ and $OC_{o,nf}$), 65 ± 4% is derived from non-fossil sources throughout of the year, with decreased contribution during wintertime (~60%). Using multiple state-of-the-art analytical techniques (e.g., $^{14}C$ measurements and aerosol mass spectrometry), R. J. Huang et al. (2014) found higher non-fossil contribution to SOC (65–85%) in Xi'an and Guangzhou and lower non-fossil contribution to SOC (35–55%) in Beijing and Shanghai in winter 2013.

These findings underline the importance of the non-fossil contribution to SOC formation in Chinese megacities. The considerable differences in SOC composition in different cities might be due to the significant difference in SOC precursors from different emission sources and atmospheric processes.



### 3.5 Fossil WIOC vs. fossil EC

Figure 7a shows a scatter plot of $WIOC_{fossil}$ and $EC_{fossil}$ concentrations. $EC_{fossil}$ is emitted by the combustion of fossil fuels, mainly coal combustion and vehicle emissions in Xi'an. $WIOC_{fossil}$ increases concurrently with $EC_{fossil}$ suggests that primary emissions by fossil fuel combustion are an important source for $WIOC_{fossil}$ as well. However, a much higher slope of

$WIOC_{fossil}$ against $EC_{fossil}$ was found in winter when compared with warm periods, implying that $WIOC_{fossil}$ and $EC_{fossil}$ originated from different fossil sources in winter and warm periods. In northern China, coal is used widely in winter for heating, which has higher primary OC/EC ratios than vehicle emissions.

The ratio of $WIOC_{fossil}$ to $EC_{fossil}$ ($(WIOC/EC)_{fossil}$) can give real world constraints on primary OC/EC ratios of an integrated fossil source. In the warm period, individual $(WIOC/EC)_{fossil}$ measured in this study ranged from 0.62 to 1.1 (averaged 0.85 ±

0.14), falling into the range of typical primary OC/EC ratios for vehicle emissions in tunnel studies (Cheng et al., 2010; Dai et al., 2015; Cui et al., 2016), excluding sample Summer-L with the highest $(WIOC/EC)_{fossil}$ ratio of 1.4 (Fig. 7b). The higher $(WIOC/EC)_{fossil}$ for Summer-L is likely due to the less efficient removal of WIOC in cleaner periods in contrast to more polluted periods during summertime. The more stagnant conditions in more polluted periods (Fig. 3) provide longer time for photochemical processes and SOC formation contributing formation of WSOC and result in decreased $(WIOC/EC)_{fossil}$ ratios

as discussed in Sect. 3.2. The $(WIOC/EC)_{fossil}$ during wintertime averaged 1.6 ± 0.1, which is closer to the primary OC/EC ratios for coal combustion than that for vehicle emissions (Fig. 7b), suggesting coal combustion is an important fossil source in winter besides vehicle emissions. Higher $(WIOC/EC)_{fossil}$ ratios in winter than in the warm period is also found in Beijing in northern China, with $(WIOC/EC)_{fossil}$ ratio of 1.6–2.4 in winter versus 0.7–1.2 in the warm period (Liu et al., 2018). However, no strong seasonal trends of $(WIOC/EC)_{fossil}$ ratios was found in southern Chinese cities, such as Shanghai (range:

1.2–1.6; Liu et al., 2018), Guangzhou (range: 0.7–1.4; Liu et al., 2018) and Hainan (around 1; Y. L. Zhang et al., 2014). Lower $(WIOC/EC)_{fossil}$ ratios were found in the Netherlands (0.6 ± 0.3; Dusek et al., 2017), Switzerland or Sweden (ranging roughly from 0.5 to 1; Szidat et al., 2004, 2009). Those higher values in China than in Europe could be attributed to the combined effects of less efficient combustion of fuel in older vehicles in China and higher primary OC/EC ratios from coal combustion that is more common in China (especially in winter in northern China) than in Europe.

In warm period, most of individual $(WIOC/EC)_{fossil}$ falls in the range of primary OC/EC ratio for vehicle emissions, indicating that vehicle emission is the overwhelming fossil source with negligible contribution from coal combustion. However, EC source apportionment by combing $F^{14}C$ and $\delta^{13}C$ of EC in this study (Fig. 5) and previous studies in Xi'an (Wang et al., 2015; Ni et al., 2018) indicates that even in the warm period, coal combustion is also an important source of fine particles. Another inconsistency is that the considerable difference in $(WIOC/EC)_{fossil}$ between winter and warm period

suggests strong seasonal variation of  coal combustion, whereas only moderate seasonal changes of $\delta^{13}C_{EC}$ were observed. Those contradictions will be discussed in the following section.





**3.6 Fossil OC: water-insoluble OC versus primary OC, water-soluble OC versus secondary OC**

Fossil WIOC (WIOC$_{fossil}$) and WSOC (WSOC$_{fossil}$) has been used widely as proxies of the fossil POC (POC$_{fossil}$) and SOC (SOC$_{fossil}$), respectively (e.g., Liu et al., 2014; Y. L. Zhang et al., 2014), because primary OC from fossil sources are mainly WIOC. Figure 8 compares the mass concentrations of WIOC$_{fossil}$ with POC$_{fossil}$, as well as WSOC$_{fossil}$ with SOC$_{fossil}$. The

wider uncertainty ranges of POC$_{fossil}$ and SOC$_{fossil}$ than $^{14}$C-apportioned WIOC$_{fossil}$ and WSOC$_{fossil}$ are mainly propagated from wide range of primary OC/EC ratios for fossil emissions (Sect. 2.5).

The same trend is observed for WIOC$_{fossil}$ and POC$_{fossil}$ throughout the year (Fig. 8a). In winter, the averaged WIOC$_{fossil}$ concentrations of 7.1 ± 3.5 µg m$^{-3}$ (± SD) matched the averaged POC$_{fossil}$ concentrations of 6.0 ± 3.3 µg m$^{-3}$. However, in the warm period, the WIOC$_{fossil}$ concentrations (1.8 ± 1.4 µg m$^{-3}$) do not match the estimated POC$_{fossil}$ (2.7 ± 2.0 µg m$^{-3}$) equally

well. WIOC$_{fossil}$ is still highly correlated with POC$_{fossil}$ but deviates strongly from the 1:1 line of WIOC$_{fossil}$ against POC$_{fossil}$, with a linear regression having a slope of 1.31, and intercept of 0.32 and an $R^2$ of 0.92. The higher POC$_{fossil}$ than WIOC$_{fossil}$ is well outside the measurement uncertainties, at least for most of samples representing high (H) and medium (M) TC concentrations (i.e., Spring-H, Spring-M, Summer-H, Autumn-H and Autumn-M). Previous studies have found that a part of WIOC can also be secondary origin from fossil sources in Egypt (Favez et al., 2008), France (Sciare et al., 2011) and Beijing,

China (Zhang et al., 2018), but this would cause the opposite trend (higher WIOC$_{fossil}$ than POC$_{fossil}$). The best explanation for the differences in summer and spring during polluted periods is the loss of fossil WIOC, indicated by decreased (WIOC/EC)$_{fossil}$ when pollution gets worse. This is probably due to more stagnant conditions in polluted periods, which allows for accumulation of pollutants and also more time for photochemical processing of WIOC and SOC formation, as discussed in Sect. 3.2. Evaporation of WIOC is not a likely explanation for this trend as temperatures do not differ strongly

between clean and polluted periods and partitioning to the gas-phase should be stronger in clean conditions. However, this decreasing trend of (WIOC/EC)$_{fossil}$ with increasing TC is not found in autumn, where WIOC$_{fossil}$ is lower than estimated POC$_{fossil}$ by a roughly constant factor. In the fall wind speed is generally low and not very variable, and photochemical processing would be weaker than in the summer and spring.

Overall, the most likely explanation for the difference between WIOC$_{fossil}$ and POC$_{fossil}$ is the overestimate of POC$_{fossil}$ by the

EC tracer method. POC$_{fossil}$ is calculated by multiplying EC$_{fossil}$ with primary OC/EC ratios for fossil sources ($r_{fossil}$ in Eq. 11). Thus, an overestimate of POC$_{fossil}$ result have two causes. First, $r_{fossil}$ might be overestimated (as EC$_{fossil}$ is well constrained by $^{14}$C), which could result either from a too high estimated fraction of coal burning in the warm period, or through rapid evaporation of POC at warmer temperatures. In the warm period, semi-volatile OC from fossil emission sources partitions more readily to the gas-phase leading to lower primary OC/EC ratios compared to winter. This is supported by laboratory

studies and ambient observations, which find that the primary OC/EC ratio for vehicle emissions is lower in warm period than in winter (Xie et al., 2017; X. H. H. Huang et al., 2014). Second, during longer residence time in the atmosphere POC might not be chemically stable and $r_{fossil}$ decreases with aging time in the atmosphere. This is the only mechanism that can





explain the decreasing $WIOC_{fossil}/EC_{fossil}$ ratios with higher pollutant concentrations and it is in line with findings from our earlier study that OC loss due to active photochemistry is more intense under high temperature and humidity in a warm period than in a cold winter (Ni et al., 2018).

As a consequence, a good match between $WSOC_{fossil}$ and $SOC_{fossil}$ was observed in winter. As shown in Fig. 8d, the 3 data

points fall close the 1:1 line of $WSOC_{fossil}$ against $SOC_{fossil}$. However, in the warm period, the data points fall below the 1:1 line of $WSOC_{fossil}$ against $SOC_{fossil}$, with a linear regression having a slope of 0.62, and intercept of 0.01 and an $R^2$ of 0.92. Higher $WSOC_{fossil}$ than $SOC_{fossil}$ can be explained by either underestimated $SOC_{fossil}$ or overestimated $WSOC_{fossil}$, or both. $SOC_{fossil}$ is calculated by subtracting $POC_{fossil}$ from $OC_{fossil}$. Thus, underestimated $SOC_{fossil}$ in warm period can result directly from overestimated $POC_{fossil}$ due to active OC loss. On the other hand, measurements of fresh emissions from fossil sources

show that a small fraction of primary fossil OC is water-soluble (Dai et al., 2015; Yan et al., 2017). If the differences between $WSOC_{fossil}$ and $SOC_{fossil}$ are considered as the primary $WSOC_{fossil}$, the primary $WSOC_{fossil}$ would constitute 25–55% $POC_{fossil}$, which is much larger than that observed in fresh fossil emissions (< 10 %; Dai et al., 2015; Yan et al., 2017). Thus, the small fraction of WSOC in primary fossil OC is not enough to explain the differences between $WSOC_{fossil}$ and estimated $SOC_{fossil}$.

The comparisons between $WIOC_{fossil}$ and $POC_{fossil}$, $WSOC_{fossil}$ and $SOC_{fossil}$ suggest that it is feasible to use $WIOC_{fossil}$ and $WSOC_{fossil}$ as indicator of $POC_{fossil}$ and $SOC_{fossil}$, respectively, with respect to trends and variations of $POC_{fossil}$ and $SOC_{fossil}$. However, the absolute concentrations of $WIOC_{fossil}$ and $WSOC_{fossil}$ are not equal to those of respective estimated $POC_{fossil}$ and $SOC_{fossil}$, especially in the warm period. If we consider photochemical loss as the primary reason of the differences between $WIOC_{fossil}$ and $POC_{fossil}$, $WSOC_{fossil}$ and $SOC_{fossil}$, then [14]C-based $WIOC_{fossil}$ and $WSOC_{fossil}$ are probably a better

approximation for primary and secondary fossil OC, respectively, than $POC_{fossil}$ and $SOC_{fossil}$ estimated using the EC tracer method (Sect. 2.5, Eqs. 7–10).

## 4 Conclusions

This study presents the first 1-year source apportionment of various carbonaceous aerosol fraction, including EC, OC, WIOC and WSOC in Xi'an, China based on radiocarbon ([14]C) measurement for the year 2015/2016. [14]C analysis shows that non-

fossil sources are an important contributor to OC fractions throughout the year, accounting for $58 \pm 6\%$ WSOC, $53 \pm 4\%$ OC and $55 \pm 5\%$ WIOC, whereas fossil sources dominated EC, with non-fossil sources contributing $18 \pm 6\%$ EC on the yearly average. An increased contributions of non-fossil sources to all carbon fractions were observed in winter, because of enhanced non-fossil activities in winter, mainly biomass burning. Fossil sources of EC were further divided into liquid fossil fuel combustion (i.e., vehicle emissions) and coal combustion by combining radiocarbon and stable carbon signatures in a

Bayesian Markov chain Monte Carlo (MCMC) approach. The MCMC results indicate that liquid fossil fuel combustion dominated EC over the whole year, contributing more than half of EC in the warm period and ~46% of EC in winter, despite



the source changes in different seasons. The remaining fossil EC was contributed by coal combustion: in winter, coal combustion (~25%) and biomass burning (~28%) equally affected EC, whereas in the warm period, coal combustion contributed a larger fraction of EC than biomass burning did.

5 Concentrations of all carbon fractions were higher in winter than in the warm period. Non-fossil WSOC was responsible for ~35% of the increased OC mass in winter, followed by non-fossil WIOC (~24%), fossil WIOC (~ 22%; $WIOC_{fossil}$) and fossil WSOC (~ 19%; $WSOC_{fossil}$). Fossil EC and biomass burning EC on average accounted for 62 % and 38 % increased EC mass in winter. Fossil WIOC/EC ratios (($WIOC/EC)_{fossil}$) in the warm period averaged $0.85 \pm 0.14$, well within the range of typical primary OC/EC ratios for vehicle emissions in tunnel studies (Cheng et al., 2010; Dai et al., 2015; Cui et al., 2016). Much higher $(WIOC/EC)_{fossil}$ values were found in winter, with an average of $1.6 \pm 0.11$, which is closer to the primary 10 OC/EC ratios for coal combustion ($2.38 \pm 0.44$; Sect. 2.5) than that for vehicle emissions, indicating additional contribution from coal burning in winter. Higher $(WIOC/EC)_{fossil}$ in winter than in the warm period is also found in Beijing in northern China (Liu et al., 2018). However, no strong seasonal trends of $(WIOC/EC)_{fossil}$ was found in southern China, such as Shanghai (Liu et al., 2018), Guangzhou (Liu et al., 2018) and Hainan (Y. L. Zhang et al., 2014), where there is no official heating season using coal.

15 The majority (60–76%) of the non-fossil OC was water-soluble in all seasons, probably resulting from the mostly water-soluble biomass-burning POC and SOC and biogenic SOC. The fossil OC in winter is less water-soluble than warm period, suggesting an enhanced SOC formation from fossil VOCs from vehicle emissions and/or coal burning in the warm period. In spring and summer, there is a clear increasing trend of $(WSOC/OC)_{fossil}$ and decreasing trend of $(WIOC/EC)_{fossil}$ in more polluted conditions. This suggests that the fossil WSOC formation as well as fossil WIOC removal increase under the 20 stagnant conditions that characterize polluted periods and allow for accumulation of pollutants and also photochemical processing and secondary OC formation. $WIOC_{fossil}$ and $WSOC_{fossil}$ have been used widely as proxies of the fossil POC ($POC_{fossil}$) and SOC ($SOC_{fossil}$), respectively, since primary fossil sources tend to produce mainly WIOC. In winter, mass concentrations of $WIOC_{fossil}$ were comparable to $POC_{fossil}$ and $WSOC_{fossil}$ to $SOC_{fossil}$, where $POC_{fossil}$ and $SOC_{fossil}$ are estimated using EC tracer method. However, the agreement was worse in the warm period, even though the respective 25 concentrations were highly correlated. This indicates that it is feasible to use $WIOC_{fossil}$ and $WSOC_{fossil}$ as indicator of $POC_{fossil}$ and $SOC_{fossil}$, respectively, with respect to trends and variations of $POC_{fossil}$ and $SOC_{fossil}$. However, the absolute concentrations of $WIOC_{fossil}$ and $WSOC_{fossil}$ are not equal to those of estimated $POC_{fossil}$ and $SOC_{fossil}$, especially in the warm period. The higher mass of $POC_{fossil}$ than $WIOC_{fossil}$ in the warm period was probably due to overestimated $POC_{fossil}$ (thus underestimated $SOC_{fossil}$) resulted from overestimated primary fossil OC/EC ratios. In the warm period, at relatively high 30 temperatures, semi-volatile OC from emission sources becomes volatilized more quickly owing to higher temperatures, leading to lower primary OC/EC ratios than other seasons. This is in line with the laboratory and ambient observations that the primary OC/EC ratio for vehicle emissions is lower in the warm period than in winter (Xie et al., 2017; X. H. H. Huang



et al., 2014), and the findings from our earlier study that in the warm period, that photochemical OC loss is active and affect final OC concentrations (Ni et al., 2018). We suggest that WIOC$_{fossil}$ and WSOC$_{fossil}$ are probably a better approximation for primary and secondary fossil OC, respectively, than POC$_{fossil}$ and SOC$_{fossil}$ estimated using the EC tracer method.

**Data availability**

All data needed to evaluate the conclusions in this study are present in the paper and the Supplment. Additional data related to this paper are availabe upon request to the corresponding author.

**Author contributions**

UD, RJH, HN and JC designed the study. HN and HD conducted the [14]C measurements. HN, HD and UD interpreted the [14]C data. JG preformed the measurements of stable isotope [13]C. HN and UD interpreted the [13]C data. HN and UD prepared

display items and HN wrote the mansucript. All authors commented on and discussed the manuscript.

**Competing interests**

The authors declare that they have no conflict of interest.

**Acknowledgments**

This work was supported by the National Key Research and Development Program of China (no. 2017YFC0212701), the

National Natural Science Foundation of China (NSFC; no. 91644219 and 41877408), and a KNAW project (no. 530-5CDP30). The authors acknowledge the financial support from the Gratama Foundation. Special thanks are given to Dipayan Paul, Marc Bleeker and Henk Been for their help with the AMS measurements at CIO, and to Anita Aerts-Bijma and Dicky van Zonneveld for her help with [14]C data correction at CIO.



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



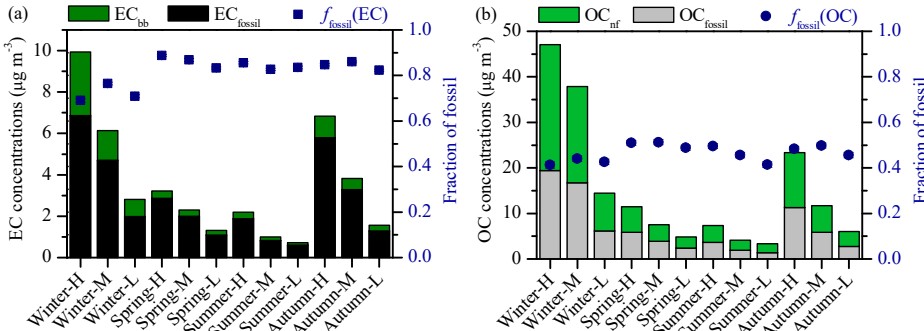

**Figure 1. (a)** Mass concentrations of EC from fossil and non-fossil sources (EC$_{fossil}$ and EC$_{bb}$, respectively), and fraction of fossil in EC ($f_{fossil}$(EC)). **(b)** Mass concentrations of OC from fossil and non-fossil sources (OC$_{fossil}$ and OC$_{nf}$, respectively), and fraction of fossil in OC ($f_{fossil}$(OC)).





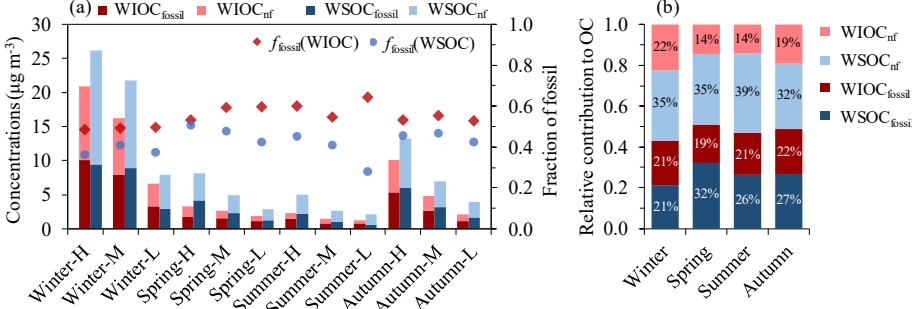

**Figure 2. (a)** Mass concentrations of WIOC and WSOC from fossil and non-fossil sources (WIOC$_{fossil}$, WIOC$_{nf}$, WSOC$_{fossil}$ and WSOC$_{nf}$) as well as fraction of fossil in WIOC and WSOC ($f_{fossil}$(WIOC) and $f_{fossil}$(WSOC), respectively). **(b)** Averaged relative contribution to OC (%) from WIOC$_{nf}$, WSOC$_{nf}$, WIOC$_{fossil}$, and WSOC$_{fossil}$ in each season.



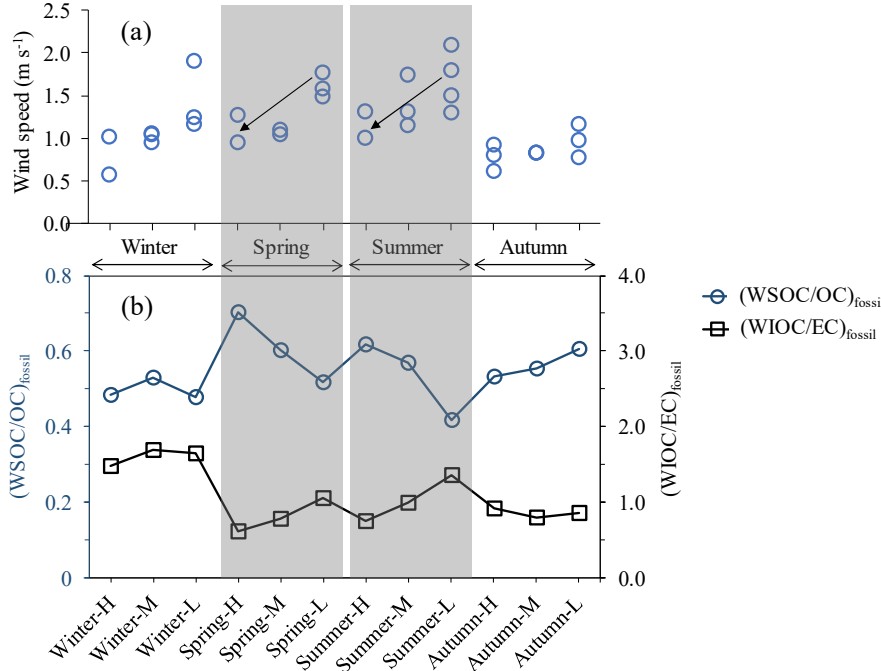

**Figure 3. (a)** Wind speed for each composite sample. Each composite sample consists of 2–4 24h filter samples, and each filter sample is shown as individual datapoint. The wind speed is recorded by the Meteorological Institute of Shaanxi Province, Xi'an, China. **(b)** The fraction of fossil WSOC in fossil OC ((WSOC/OC)$_{fossil}$, dark blue circle), the fossil WIOC to fossil EC ratio ((WIOC/EC)$_{fossil}$, black square) over all the selected samples throughout the year.



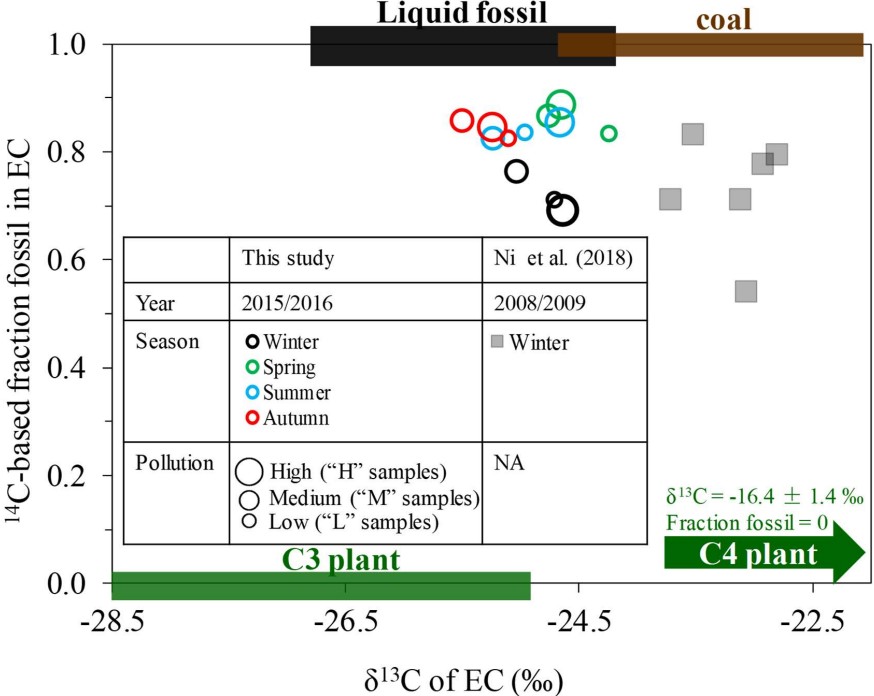

**Figure 4.** The $^{14}$C-based fraction fossil versus $\delta^{13}$C for EC in Xi'an, China in different seasons in 2015/2016 (this study, circle symbols), compared with those in winter 2008/2009 from Ni et al. (2018) (square symbols). The size of the symbols for the year 2015/2016 (this study) represents the pollution conditions (high, medium and low) for each sample. The expected $^{14}$C and $\delta^{13}$C endmember ranges for

5   emissions from C3 plant burning, liquid fossil fuel burning and coal burning are shown as green, black and brown bars, respectively. The $\delta^{13}$C signatures are indicated as mean ± SD (Sect. 2.6). The $\delta^{13}$C signatures of corn stalk (i.e., C4 plant) burning is -16.4 ± 1.4 ‰ is also indicated.





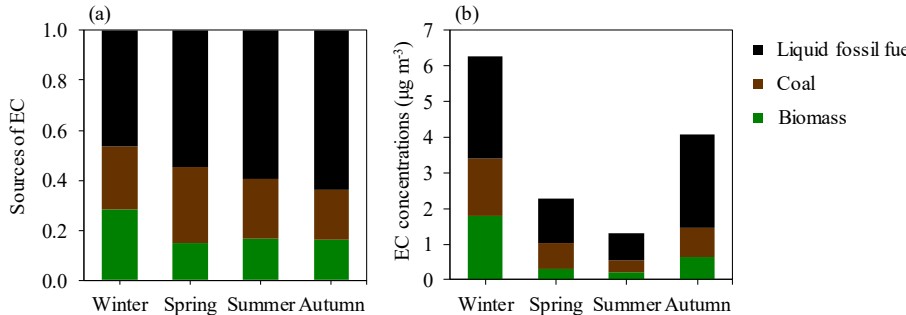

**Figure 5. (a)** Fractional contributions of 3 incomplete combustion sources to EC in different seasons. **(b)** Mass concentration of EC (μg m⁻³) from each combustion source. The data are presented in Tables S8 and S9.

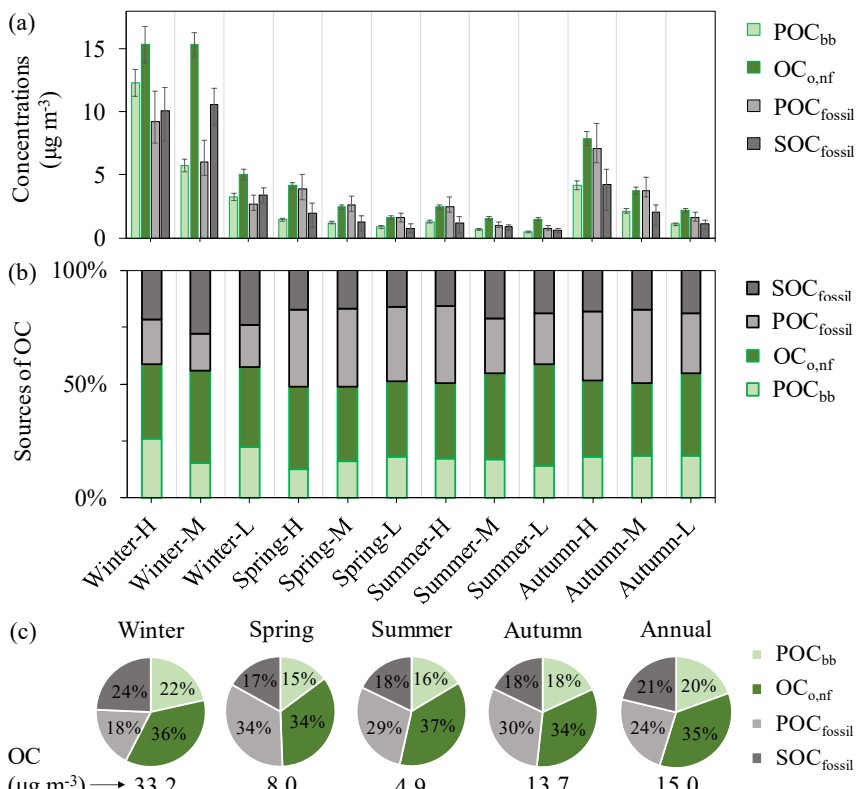

**Figure 6.** **(a)** The estimated mass concentrations of $POC_{bb}$, $OC_{o.nf}$, $POC_{fossil}$, $SOC_{fossil}$ (µg m$^{-3}$) in total OC of PM$_{2.5}$ samples. The error bars indicate the interquartile range (25th–75th percentile) of the median values. **(b)** The percentage of $POC_{bb}$, $OC_{o.nf}$, $POC_{fossil}$, $SOC_{fossil}$ in total OC. **(c)** Average source apportionment results of OC in each season and over the year. The numbers below the pie charts represent the seasonally/annually averaged OC concentrations.





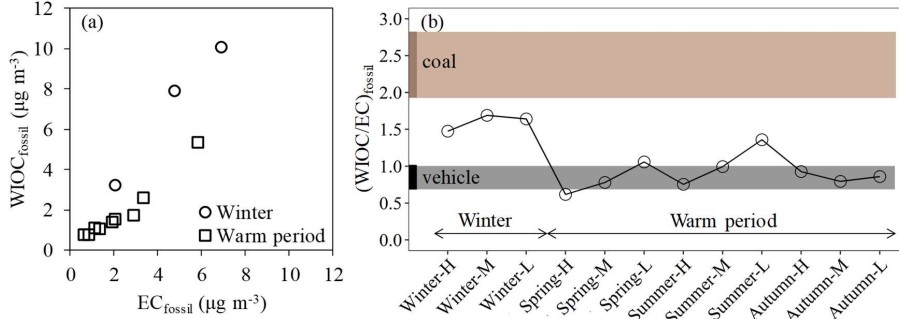

**Figure 7. (a)** A scatter plot of EC concentrations from fossil sources ($EC_{fossil}$) versus WIOC concentrations from fossil sources ($WIOC_{fossil}$) in winter (circle) and warm period (square). **(b)** The WIOC to EC ratio from fossil sources (($WIOC/EC)_{fossil}$) over all the selected samples throughout the year. The dashed areas indicate typical primary OC/EC ratios for coal combustion (brown) and vehicle emissions (black).





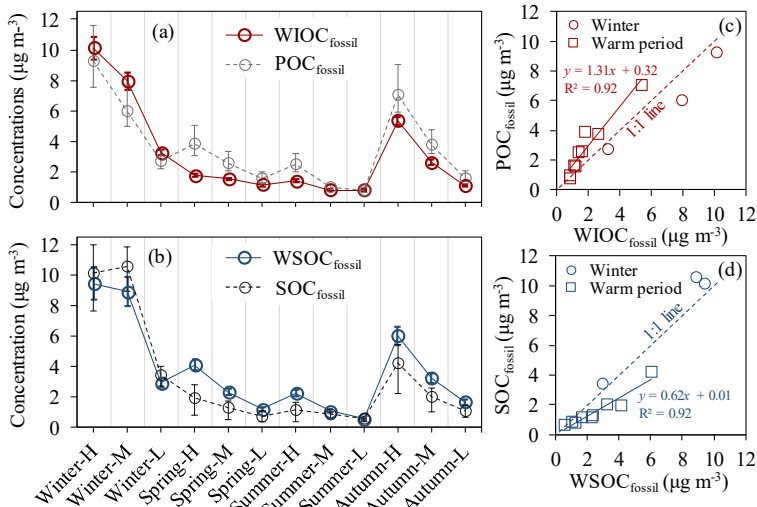

**Figure 8. (a)** Concentrations of WIOC and POC from fossil sources (WIOC$_{fossil}$ and POC$_{fossil}$, respectively)**.** Panel a has the same *x* axis with panel b. **(b)** Concentrations of WSOC and SOC from fossil sources (WSOC$_{fossil}$ and SOC$_{fossil}$, respectively)**. (c)** A scatter plot of WIOC$_{fossil}$ concentrations versus POC$_{fossil}$ concentrations. **(d)** A scatter plot of WSOC$_{fossil}$ concentrations versus SOC$_{fossil}$ concentrations. The interquartile range (25th-75th percentile) of the median POC$_{fossil}$ and SOC$_{fossil}$ is shown by grey vertical bars in panel a and black vertical bars in panel b.