# Peer review of "Sources and formation of carbonaceous aerosols in Xi'an, China: primary emissions and secondary formation constrained by radiocarbon"

_Atmospheric Chemistry and Physics, 2019_

## Referee Comment (RC1) · Anonymous Referee #2 · 30 Jul 2019

The paper acp-2019-437 by Ni et al. deals with carbon isotope measurements (14C and $\delta$13C) measurements on carbon fractions carried out in China. The analysed samples cover 33 days throughout one year, covering all seasons and low, medium, high concentrations. The paper is clear, generally well written and the presented data are of interest for the scientific community and for future development of efficient abatement strategies. 14C data on carbon fractions are still relatively rare due to particular treatment of the sample and the need of accelerator mass spectrometry for isotopic ratio quantification. Nevertheless, few major concerns should be solved before the paper can be published on ACP.

[Figure]

Major concerns Pag.3 line 17: "1-year 14C measurements". From this sentence, I would expect high percentage of day coverage throughout the year. Opposite, Figure S1 evidences that 33 days are covered (less than 10% day coverage). The reviewer is aware of the difficulties related to 14C measurements and appreciates the efforts to make the analyses representative of all seasons and aerosol loadings. Nevertheless, the sentence is somewhat misleading. Please rephrase.

Paragraph 2.2: information on field blanks is completely missing and should be added

Pag.4, line 21-22: "Extraction of EC was done by heating the carbon that remained on the filters at 850 °C for 5 h". In air or oxygen? Could you provide information on EC recovery for this kind of analysis (e.g. compared to EC quantification by TOT?). Is it similar to the one for 14C analysis?

Pag.6, line 24: "Currently, the F14C of the atmospheric $CO_2$ is approximately 1.04 (Levin et al., 2008)". Why do not using more updated values? (see e.g. https://www.atmos-chem-phys.net/18/6187/2018/acp-18-6187-2018.pdf)

Pag.7, line 15: "F14Cbb = 1.10 ± 0.05". Please clarify assumptions on wood age and fell date.

Pag.7, line 18: "F14Cnf =1.09 ± 0.05": it seems to be fully dominated by wood burning. Please, clarify how it was obtained.

Pag.9, line 10-17: overlapping interval for expected $\delta$13C for nearly all sources is present. This makes the analysis very weak, also considering that results are in contrast with 14C results (see pag.15, last paragraph). The reason to maintain this section and the related analyses should be better clarified. Figure S4 should be added to the text as it evidences the difficulties in apportioning coal and liquid fossil fuel contributions separately

Pag.10, line 16: "slight but consistent tendency": what is "slight"? And in what sense "consistent"? The authors should specify the statistical approach used to verify "consistency".

Page 10, line 22: "lower in other seasons (around 15%) with a slightly lower values in spring ($14 \pm 3\%$)". Is spring really different compared to autumn and summer? As it is mentioned, it should be proved by statistical tests)

Page 10, Lines 21-29: table S1 merits to be added to the manuscript, as not all the numbers are reported in the text.

Page 11, Line 5: "$6.8 \pm 6.0\ \mu$g m-3". Maybe interquartile range is more significant than standard deviation, as the data distribution is not expected to follow a gaussian curve. Same comment for analogous representation of absolute concentrations in the rest of the text (e.g. pag.11, lines 6, 22)

Pag.11, line 21-23: "larger than", "comparable with": which are the statistical criteria used to evaluate comparability?

Pag.12, line 13: "The fossil OC is less water soluble in winter with lower (WSOC/OC)fossil ratios of around 0.5 than in the warm period". What is "warm period"? Why indicating the value during winter and not during the warm period? Are the differences statistically significant, also considering the limited number of data available?

Paragraph 3.3: similarities in $\delta$13C reference values for different sources affect the results presented here. The results show very high variability and this should be better commented in the text, also in the light of figure S4.

Pag.13 line 31: "moderately". Quantify and evaluate statistical significance

Pag.14, line 1: "more constant". Compared to what?

Pag.14, line 24: "rapid". Please quantify (hours? Days?) and justify the sentence.

Pag.15, line 31: "Those contradictions will be discussed in the following section". Coal is hardly mentioned in the following paragraph, thus it is unclear what the authors are

referring to.

Pag.16, line 11 and Pag.17, line 6: "slope of 1.31, and intercept of 0.32 and an R2 of 0.92". "a slope of 0.62, and intercept of 0.01 and an R2 of 0.92". As important uncertainties affect quantities both on x and y axis, 2-sided (Deming) regression should be attempted for better representation of these regression lines

Pag.17, line 10. "that a small fraction of primary fossil OC is water-soluble (Dai et al., 2015; Yan et al., 2017).". This sentence should be moved more above, as it is also a justification of higher fossil POA compared to fossil WIOC.

Pag.19, line 2: "We suggest that WIOCfossil and WSOCfossil are probably a better approximation for primary and secondary fossil OC, respectively, than POCfossil and SOCfossil estimated using the EC tracer method". This is in contrast with the sentence at the previous point.

Minor comments Page 4, line 15: "< 0.2 $\mu$g m-2) compared to the TC loading of the samples (13–246 $\mu$g m-2". Replace with "< 0.2 $\mu$g cm-2) compared to the TC loading of the samples (13–246 $\mu$g cm-2"

Pag.5 line 7: "WSOC can be calculated as the difference between OC and WIOC". Unclear why this sentence is here. The previous reference to radiocarbon measurements is confusing (as radiocarbon determination is not carried out as difference, as explained on page 7)

Page 5, line 19: "By water-extraction, water-soluble OC (WSOC) is removed from filter pieces (Dusek et al., 2014)". The role of WSOC removal as a key procedure for reducing the impact of possible pyrolysis on 14C measurements of EC merits to be better evidenced as a key step for the correct 14C in EC measurement. In the years 2012-2014 three thermal treatments were developed nearly in parallel and all of them identified WSOC removal as a key step for radiocarbon measurement on EC. Suitable reference should include also Zhang et al, 2012 (https://doi.org/10.5194/acp-12-10841-2012) and

Bernardoni et al, 2013 (http://dx.doi.org/10.1016/j.jaerosci.2012.06.001). Please note that these were the methods object of the inter-comparison reported in the mentioned Zenker et al., 2017 papers.

Page 10, lines 23-24: "Beijing shows a very different seasonal trend, where fbb(EC) was lowest in summer (∼7%) and increased to ∼20% during the rest of the year (Zhang et al., 2017)". Please, introduce the sentence, e.g. "By comparison with literature data for Beijing" Page 10 line 30 (and following): change "around" with "about"

Pag.16, line 1: "Fossil WIOC (WIOCfossil) and WSOC (WSOCfossil) has been used". Change into "Fossil WIOC (WIOCfossil) and WSOC (WSOCfossil) have been used"

Pag.16, line 28: "Thus, an overestimate of POCfossil result have two causes". Change into: "Thus, an overestimate of POCfossil result has two causes.

Pag. 17, line 27: "An increased contributions". Change into: "An increased contribution"

---

## Referee Comment (RC2) · Anonymous Referee #3 · 23 Aug 2019

The paper reports results of a 1-year source apportionment of carbonaceous aerosol fractions in a polluted Chinese city, based on radiocarbon ($^{14}$C) and stable carbon isotope($\delta^{13}$C) measurements. Large focus is devoted to $^{14}$C source apportionment of WIOC and WSOC, and discussion on whether $^{14}$C-apportioned WIOC and WSOC can be used as proxies of primary emissions and secondary formation of OC, respectively. To my knowledge, there are limited $^{14}$C results of WIOC and WSOC in the literature, especially this study covers a full year cycle. The data and methodology are presented clearly and appear to be valid. The well-writen manuscript is acceptable for publication after minor revisions.

1) $^{14}$C measurement is known to be expensive and time-comsuming. In this study, only 3 samples/season (in total, 12 samples/year) were selected for $^{14}$C measurements of EC, OC and WIOC. How are those 12 samples representative of a year?

2) Section 2.2. Are the samples corrected for field blanks?

3) Page 6, line 6-8. Why $\delta^{13}$C of -25‰ is used to correct isotope fractionation?

4) Page 7, line 2. "$M_{OC}$ is measured by the thermal-optical method as described in Sect. 2.2".

In Sect. 2.2, EUSAAR_2 protocol is used. In Sect.2.4.2, for $^{14}$C measurement, OC is extracted by heating filter samples in $O_2$ at 375 ℃. So I see two different protocols. How comparable are they?

5) Page 7, line 10-11. "The most likely value of $M_{WIOC}$ is chosen at $M1_{WIOC} + 2/3 \times (M2_{WIOC} - M1_{WIOC})$, because it is more likely that WIOC has a similar recovery as OC rather than 100% recovery". Do you have any evidence to support this statement? I care this because the estimated $M_{WIOC}$ is used in Eq. 4 to determine the $F^{14}$C and mass of WSOC.

6) Page 7, line 14-20. Conversion factors are applied to convert $F^{14}$C to the relative contribution of non-fossil sources to EC/OC. The conversion factors are $F^{14}C_{bb}$ (= 1.10 ±0.05) for EC and $F^{14}C_{nf}$ (= 1.09 ±0.05) for OC, respectively. Why are the two conversion factors slightly different? I suggest the authors to explain this clearly in the method section.

7) Page 9, line 15-16. Are the measurement uncertainties of $F^{14}C_{(EC)}$ and $\delta^{13}C_{EC}$ considered in the MCMC calculations?

**Technical comments:**

8) Page 4, line 1. "a" between "in" and "pre-baked" should be deleted.

9) Page 8, line 8. To be consistent with the text, I think it should a comma in "$OC_{o,nf}$" in Eq. (8).

Please check all instances

10). Page 8, line 11. A citation is missing for the statement that "In most cases, contributions of primary biogenic OC to $PM_{2.5}$ are likely small".

11) Page 8, line 19. It should be "**combining**" instead of "combing".

12) Page 8, line 25. Give full name of PDF, because it is used for the first time in this manuscript. The authors should check the manuscript again for proper use of abbreviations.

13) Page 10, line 22. "a slightly lower **value**" instead of "a slightly lower values"

14) Page 17, line 23. "various carbonaceous aerosol **fractions**"

15) Page 17, line 27. "An increased **contribution** of non-fossil sources to all carbon fractions **was** observed"

---

## Author Comment (AC1) · 15 Oct 2019

We thank the reviewers for the helpful comments and providing us the opportunity to strengthen our research. We try to address all of them carefully. Below are point-to-point responses.

**Anonymous reviewer #2:**

The paper acp-2019-437 by Ni et al. deals with carbon isotope measurements ($^{14}$C and $\delta^{13}$C) measurements on carbon fractions carried out in China. The analysed samples cover 33 days throughout one year, covering all seasons and low, medium, high concentrations. The paper is clear, generally well written and the presented data are of interest for the scientific community and for future development of efficient abatement strategies. $^{14}$C data on carbon fractions are still relatively rare due to particular treatment of the sample and the need of accelerator mass spectrometry for isotopic ratio quantification. Nevertheless, few major concerns should be solved before the paper can be published on ACP.

**Main comments:**

**1)** Pag.3 line 17: "1-year $^{14}$C measurements". From this sentence, I would expect high percentage of day coverage throughout the year. Opposite, Figure S1 evidences that 33 days are covered (less than 10% day coverage). The reviewer is aware of the difficulties related to $^{14}$C measurements and appreciates the efforts to make the analyses representative of all seasons and aerosol loadings. Nevertheless, the sentence is somewhat misleading. Please rephrase.

**Response:** Thank you for this valuable feedback. Following the reviewer's suggestion, we avoid using the expression "*1-year* $^{14}$C measurements" and "*1-year* source apportionment" in the revised manuscript. We thus have rephased the text (changes are underlined) to avoid misleading the reader:

> "We present, to our best knowledge, the first $^{14}$C measurements covering all four seasons that distinguish fossil and non-fossil contributions to various carbon fractions, including EC, OC, WIOC and WSOC in Xi'an." (page 3, line 17-19)

> "This study presents the first source apportionment of various carbonaceous aerosol fractions, including EC, OC, WIOC and WSOC in Xi'an, China based on radiocarbon ($^{14}$C) measurement in four seasons for the year 2015/2016." (page 19, line 7-8)

**2)** Paragraph 2.2: information on field blanks is completely missing and should be added.

**Response:** Thank you for pointing this out. The average field blank of OC was $0.9 \pm 0.2$ μg cm$^{-2}$ (N=6, equivalent to ~ $0.23 \pm 0.05$ μg m$^{-3}$), which was subtracted from the sample OC concentrations. EC on field blanks was in most cases below the detection level. We thus did not conduct the blank correction for EC concentrations. This underlined description is added in the Sect. 2.2 (page 4, line 17–19).

According, we add the sampling information of filed blanks in Method Sect. 2.1:

"Field blank filters were treated exactly like the sample filters, except that no air was drawn through the filter." (page 3, line 30 – page 4, line 1)

**3)** Pag.4, line 21-22: "Extraction of EC was done by heating the carbon that remained on the filters at 850 ∘C for 5 h". In air or oxygen? Could you provide information on EC recovery for this kind of analysis (e.g. compared to EC quantification by TOT?). Is it similar to the one for $^{14}$C analysis?

**Response:** Extraction of EC was done by heating the carbon that remained on the filters at 850 °C for 5 h in another vacuum-sealed quartz tube. We have added this in the revised text (page 4, line 25).

In this study, we used a two-step method (OC step: 375 °C for 3 h; EC step: 850 °C for 5 h) to isolate OC and EC for $\delta^{13}$C analysis. Our earlier study in Xi'an found that EC recovery for $\delta^{13}$C analysis (relative to EC quantified by the thermal-optical reflectance protocol IMPROVE_A; Chow et al., 2007) was on average $123 \pm 8$ %, higher than 100% (Zhao et al., 2018). The reason is that pyrolyzed OC (formed through charring during the OC removal procedure) and possibly some remaining OC compounds (e.g., high molecular weight refractory carbon) can be released at the high temperature of EC step.

The fraction of pyrolyzed OC in EC varies from sample to sample (Huang et al., 2006), the less the better for $\delta^{13}$C analysis of EC. However, using the two-step method, we can not achieve pure EC (mainly due to the inclusion of pyrolyzed OC), and the resulted $\delta^{13}$C of EC could be biased by $\delta^{13}$C of pyrolyzed OC, if the contribution from pyrolyzed OC to the isolated EC is high and $\delta^{13}$C of pyrolyzed OC is very different from $\delta^{13}$C of pure EC.

To examine the effect of pyrolyzed OC on $\delta^{13}$C of EC, a sensitivity analysis is performed. $\delta^{13}$C of pyrolyzed OC is not known, but our recent studies suggest that $\delta^{13}$C of pyrolyzed OC is not very different from $\delta^{13}C_{OC}$ (<1‰ in many cases). We thus use $\delta^{13}C_{OC}$ (measured but not presented in this study) to represent $\delta^{13}$C of pyrolyzed OC. $\delta^{13}$C of pure EC is calculated based on isotope mass balance. This analysis shows that for high contribution from pyrolyzed OC to the isolated EC of 20%, the expected difference in $\delta^{13}$C between measured EC and true EC is still <1‰. This will not significantly change any conclusions made in this study.

We add the above discussion in the Supplement S1. In the main text, we add:

"Pyrolyzed OC can be formed through charring during the OC removal procedure and is released at the high temperature of EC step. To assess the potential effect of pyrolyzed OC on the measured $\delta^{13}C_{EC}$, we conducted a sensitivity analysis based on isotope mass balance (See details in the Supplemental S1). This analysis shows that even for high contribution from pyrolyzed OC to the isolated EC of 20%, the expected difference in $\delta^{13}$C between measured EC and true EC is still <1‰." (page 5, line 4-7)

For $^{14}$C analysis of EC, a full recovery of EC cannot be achieved, because we applied an intermediate step to remove more-refractory OC and also less refractory EC, to completely

remove OC (Dusek et al., 2014). This has been explained in Sect. 2.4.2. This EC isolation method for $^{14}$C analysis was evaluated and compared to methods from two other laboratories, finding that the results of $^{14}$C measurements in EC agree well within their uncertainty estimates (Zenker et al., 2017). In this study, EC recovery after the intermediate 450 °C step was approximately 70%, estimated by comparing to the EC quantified by EUSAAR_2 protocol. This underlined sentence has been added to Sect. 2.4.2 (page 6, line 1-2).

**4)** Pag.6, line 24: "Currently, the F$^{14}$C of the atmospheric $CO_2$ is approximately 1.04 (Levin et al., 2008)". Why do not using more updated values? (see e.g. https://www.atmos-chem-phys.net/18/6187/2018/acp-18-6187-2018.pdf)

**Response:** We now specify the year (2010) for the F$^{14}$C of 1.04. This value cited at this point in the manuscript is for illustrative purpose only to give readers an idea how fast the F$^{14}$C in the atmosphere decreased since the stop of the nuclear bomb tests. For our estimate of F$^{14}$C$_{nf}$ for OC later on (Sect.2.5), we use the most recent value of 1.02, estimated by Vlachou et al. (2018) from $^{14}CO_2$ measurements in Schauinsland (Levin et al., 2010).

The revised text shows:

> "F$^{14}$C of carbon from fossil sources is 0, and carbon from non-fossil sources (or "contemporary" sources) should have F$^{14}$C of 1. But the extensive release of $^{14}$C from nuclear bomb tests in the late 1950s and early 1960s and $^{14}$C-free $CO_2$ from fossil fuel combustion has perturbed the atmospheric F$^{14}$C values significantly. The former increased the F$^{14}$C in the atmosphere by up to a factor of 2 in the northern hemisphere in the 1960s. The nuclear tests have been banned in the atmosphere, outer space and under water since 1963. Since then, the atmospheric F$^{14}$C has been slowly decreasing, as $^{14}$C is mainly taken up by the oceans and terrestrial biosphere and diluted by $^{14}$C-free $CO_2$ (Hua and Barbetti, 2004; Levin et al., 2010). In 2010, the F$^{14}$C of the atmospheric $CO_2$ is approximately 1.04 (Levin et al., 2008, 2010), whereas in 2014 it decreased to 1.02 (Vlachou et al., 2018)." (page 7, line 6-8)

**5)** Pag.7, line 15: "F$^{14}$C$_{bb}$ = 1.10 ± 0.05". Please clarify assumptions on wood age and fell date.

**Response:** For biogenic aerosols, aerosols emitted from cooking as well as annual crop, the F$^{14}$C is close to the value of current atmospheric $CO_2$. F$^{14}$C of wood burning is higher than that, because a significant fraction of carbon in the wood burned today was fixed during times when atmospheric $^{14}CO_2$ were substantially higher than today. As the reviewer points out, F$^{14}$C of wood burning depends on the age and origin of the wood. Estimates of F$^{14}$C for wood burning are based on tree-growth models (e.g., Lewis et al., 2004; Mohn et al., 2008) and found to range from 1.08 to 1.30 relating to wood age and fell date (Heal, 2014, and references therein).

We have added the following explanation to the revised text:

> "F$^{14}$C$_{bb}$ represents F$^{14}$C of biomass burning including wood burning and crop residue burning. This is because that biomass burning in Xi'an mainly includes household usage of wood and crop residues as well as open burning of crop residues. F$^{14}$C for burning of

annual crop has a similar value of current atmospheric $CO_2$. $F^{14}C$ of wood burning is higher than that and varies with the age of tree. Estimates of $F^{14}C$ for wood burning are based on tree-growth models (e.g., Lewis et al., 2004; Mohn et al., 2008) and found to range from 1.08 to 1.30 relating to wood age and fell date (Heal, 2014, and references therein). $F^{14}C_{bb}$ was estimated as $1.10 \pm 0.05$ for Xi'an in this study. The lower limit of $F^{14}C_{bb}$ corresponds to burning of young wood (5–10 years old tree harvested between 2010 and 2015) and crop residues as main sources of EC, and the upper end of $F^{14}C_{bb}$ corresponds to older wood (30–60 years old tree) combustion as the main source of EC."
(page 7, line 29 to page 8, line 7)

6) Pag.7, line 18: "$F^{14}C_{nf} = 1.09 \pm 0.05$": it seems to be fully dominated by wood burning. Please, clarify how it was obtained.

**Response:** $F^{14}C_{nf}$ is $F^{14}C$ of non-fossil sources including both biomass burning and biogenic emissions, and is calculated as

$$F^{14}C_{nf} = F^{14}C_{bb} \times p_{bb} + F^{14}C_{bio} \times p_{bio} \qquad (R1)$$

Where $F^{14}C_{bio}$ (=1.02) is the fraction of modern carbon of biogenic sources and was estimated from long-term $^{14}CO_2$ measurements at the Schauinsland background station (Levin and Hammer, 2013; Levin et al., 2010). $p_{bb}$ and $p_{bio}$ are the fraction of biomass burning and biogenic sources to the total non-fossil sources, respectively. In Xi'an, China, we assume that biogenic OC is not very important, due to strong anthropogenic sources as evidenced by concentrations of carbonaceous aerosols (e.g., annual average TC concentrations of 27 µg/m$^3$ in 2012/2013[Han et al.., 2016] and on average of 17 µg m$^{-3}$ for selected samples for $^{14}C$ analysis in this study), that are much higher than typical concentrations of secondary biogenic aerosols. We thus set $p_{bio}$ to $0.15 \pm 0.15$, and find out $p_{bio}$ has only a very little impact on $F^{14}C_{nf}$ compared to other uncertainties, e.g., an increase of $p_{bio}$ from 0.15 to 0.3 would lead to small change in the central value of $F^{14}C_{nf}$ from 1.09 to 1.08.

To clarify, the revised text shows:

"Analogously, the relative contribution of non-fossil sources to OC, WIOC and WSOC (i.e., $f_{nf}$(OC), $f_{nf}$(WIOC) and $f_{nf}$(WSOC), respectively) can be estimated from their corresponding $F^{14}C$ values and $F^{14}C_{nf}$ . $F^{14}C_{nf}$ is $F^{14}C$ of non-fossil sources including both biomass burning and biogenic sources. $F^{14}C$ of biogenic sources can be estimated from long-term $^{14}CO_2$ measurements at the Schauinsland background station (Levin and Hammer, 2013; Levin et al., 2010). In Xi'an, biogenic OC is probably not very important, as could be expected from high concentrations of carbonaceous aerosols and strong anthropogenic sources. $F^{14}C_{nf}$ is thus estimated as $1.09 \pm 0.05$ (Lewis et al., 2004; Levin et al., 2010; Y. L. Zhang et al., 2014). The central value of 1.09 corresponds to 15% contribution of biogenic OC to OC." (page 8, line 8-14).

**7)** Pag.9, line 10-17: overlapping interval for expected $\delta^{13}$C for nearly all sources is present. This makes the analysis very weak, also considering that results are in contrast with $^{14}$C results (see pag.15, last paragraph). The reason to maintain this section and the related analyses should be better clarified. Figure S4 should be added to the text as it evidences the difficulties in apportioning coal and liquid fossil fuel contributions separately.

**Response:** Following the reviewer's suggestion, we have moved the Figure S4 from the supplemental material to the main body of the manuscript, namely Figure 6. The order of figures in the main text and supplemental material is adapted accordingly.

The source endmembers for $\delta^{13}$C are less well-constrained than for $F^{14}$C, as $\delta^{13}$C varies with fuel types and combustion conditions. The $^{14}$C results can constrain fossil and biomass burning contribution to EC very well. But in this study, we can only separate fossil EC into EC from coal combustion and EC from vehicular emission by complementing $^{14}$C with $^{13}$C. The following sentences are added in the Sect. 2.6 to clarify this:

> "EC from fossil sources can be first separated from biomass burning by $F^{14}C_{(EC)}$. Further, $\delta^{13}C_{EC}$ allows separation of fossil sources into coal and liquid fossil fuel burning" (page 10, line 2-3)

In the Sect. 2.6, we further elaborate on the consequence of the overlapping $\delta^{13}$C source signatures:

"The source endmembers for $\delta^{13}$C are less well-constrained than for $F^{14}$C, as $\delta^{13}$C varies with fuel types and burning conditions. For example, the range of possible $\delta^{13}C_{liq.fossil}$ overlaps to a small extent with the range of $\delta^{13}C_{coal}$, although liquid fossil fuels are usually more depleted than coal. The MCMC technique takes into account the variability in the source signatures of $F^{14}$C and $\delta^{13}$C (Parnell et al., 2010, 2013), where $\delta^{13}$C introduces a larger uncertainty than $F^{14}$C. Uncertainties of $\delta^{13}C_{bb}$, $\delta^{13}C_{liq.fossil}$, $\delta^{13}C_{coal}$ and $F^{14}C_{bb}$ as well as the measured ambient $\delta^{13}C_{EC}$ and $F^{14}C_{(EC)}$ are propagated. The results of the MCMC calculations are the posterior PDFs for $f_{bb}$, $f_{liq.fossil}$ and $f_{coal}$. The PDFs of $f_{liq.fossil}$ and $f_{coal}$ are skewed. By contrast, the PDFs of $f_{bb}$ is symmetric as it is well-constrained by $F^{14}$C (Fig. 6). In this study, the median is used to represent the best estimate of the $f_{bb}$, $f_{liq.fossil}$ and $f_{coal}$. Uncertainties of this best estimate are expressed as an interquartile range (25th-75th percentile) of the corresponding PDFs." (page 10, line 11-20)

The reasons to maintain this Sect. 2.6 are mentioned several times in this manuscript, for example:

(a) in the introduction section

> "We present, to our best knowledge, the first $^{14}$C measurements covering all four seasons that distinguish fossil and non-fossil contributions to various carbon fractions, including EC, OC, WIOC and WSOC in Xi'an. Fossil sources of EC are further divided into coal and liquid fossil fuel combustion by complementing radiocarbon with the stable carbon isotopic signature." (page 3, line 17-20)

(b) in the result Sect. 3.3. Combustion sources apportioned by stable carbon isotopes

"Along with radiocarbon data, the stable carbon isotopic ratio of EC (denoted by $\delta^{13}C_{EC}$) provides additional insight into source apportionment of EC, especially between different type of fossil sources (i.e., coal versus liquid fossil fuel combustion)." (page 14, line 1-3)

(c) in the method Sect. 2.5, the results of MCMC calculations (i.e., Bayesian statistics combining $F^{14}C_{(EC)}$ and $\delta^{13}C_{EC}$) are used to estimate $p$ values, and subsequently $POC_{fossil}$ and $SOC_{fossil}$:

"$POC_{fossil}$ can be estimated from $EC_{fossil}$ and primary OC/EC ratio of fossil fuel combustion ($r_{fossil}$):

$$POC_{fossil} = EC_{fossil} \times r_{fossil}. \qquad (10)$$

Fossil sources in China are almost exclusively from coal combustion and vehicle emissions, thus $r_{fossil}$ can be estimated as

$$r_{fossil} = r_{coal} \times p + r_{vehicle} \times (1 - p), \qquad (11)$$

where $p$ is the relative contribution of coal combustion to fossil EC. That is, $p = EC_{coal}/EC_{fossil}$, where estimation of $EC_{coal}$ is achieved by combining $F^{14}C_{(EC)}$ and $\delta^{13}C_{EC}$ with the Bayesian calculations as described in details in the Sect. 2.6 and Supplement S2." (page 9, line 9-15)

**8)** Pag.10, line 16: "slight but consistent tendency": what is "slight"? And in what sense "consistent"? The authors should specify the statistical approach used to verify "consistency".

**Response:** "Consistent" is used to compare the seasonal patterns of $f_{fossil}(EC)$ and $f_{fossil}(OC)$. $f_{fossil}(EC)$ is higher in spring than in summer and autumn, and this is also true for $f_{fossil}(OC)$.

We thank the reviewer to point it out that "slight" may not be enough to quantify the seasonal changes, and in the revised text we add the $f_{fossil}(EC)$ and $f_{fossil}(OC)$ values in spring and in summer and autumn:

"The $f_{fossil}(EC)$ and $f_{fossil}(OC)$ follow the same seasonal trends: the values are lower in winter and higher in the rest of the seasons (i.e., warm period). Within the warm period, both are slightly higher in spring ($f_{fossil}(EC) = 86 \pm 3\%$, $f_{fossil}(OC) = 50 \pm 1\%$) than in summer and autumn ($f_{fossil}(EC) = 84 \pm 2\%$, $f_{fossil}(OC) = 47 \pm 3\%$) in general and also to be slightly lower under the cleanest periods (i.e., in spring, summer and autumn, $f_{fossil}(EC)$ and $f_{fossil}(OC)$ in polluted days ("H" and "M" samples) were higher than in clean days ("L" samples): Fig. 1, Tables 1, S5)." **(page 11, line 15-19)**

We did not perform any statistical approach to verify "slight" and "consistency". Because our sample sizes are too small to determine if a parametric test (that assume a normal distribution) is applicable. However, either for a parametric or non-parametric test the statistical power is low for small sample sizes, e.g., to compare polluted and non-polluted days we would compare samples with 6 vs 3 data points, which makes any statistical test unreliable. Therefore, we only qualitatively describe apparent trends, illustrated in Figure R1. As shown in Fig. R1(a), $f_{fossil}(OC)$ in both polluted days ("H" and "M" samples) and clean days ("L" samples) is higher in spring

than that in summer and autumn. Moreover, in spring, summer and autumn, the $f_{fossil}(OC)$ is lower in clean days than in polluted days. Those are mostly true for $f_{fossil}(EC)$ as shown in Fig. R1(b), except that $f_{fossil}(EC)$ for sample Summer-M ($0.827 \pm 0.005$) and Summer-L ($0.835 \pm 0.006$).

[Figure]

**Figure R1.** (a) $f_{fossil}(OC)$ in both polluted days ("H" and "M" samples) and clean days ("L" samples) in different seasons. (b) $f_{fossil}(EC)$ in both polluted days ("H" and "M" samples) and clean days ("L" samples) in different seasons

**9)** Page 10, line 22: "lower in other seasons (around 15%) with a slightly lower values in spring ($14 \pm 3\%$)". Is spring really different compared to autumn and summer? As it is mentioned, it should be proved by statistical tests)

**Response:** The $f_{bb}(EC)$ averages $14 \pm 3$ % ($\pm$ SD; N=3) in spring, and $16 \pm 2\%$ ($\pm$ SD; N=6) in autumn and summer. Because our sample sizes are small, it is difficult to determine if a parametric test (that assume a normal distribution) is applicable. In addition, the statistical power is low for small sample sizes. We thus decide not to perform any statistical tests to compare $f_{bb}(EC)$ in spring and in autumn and summer. In contrast, Figure R2 allow us to examine the data distribution, and we can see that in both polluted days ("H" and "M" samples) and clean days ("L" samples), $f_{bb}(EC)$ is a bit lower in spring ($\underline{14 \pm 3\%}$) than in summer and autumn ($\underline{16 \pm 3\%}$), but the trend is not clear.

To be specify, we revised the text and delete the sentence "with a slightly lower values in spring (14 ± 3%)":

"$f_{bb}(EC)$ is higher in winter (28 ± 4%) than that in other seasons (i.e., warm period, on average 15 ± 2 %)." (page 11, line 24-25)

[Figure]

**Figure R2.** $f_{bb}(EC)$ in both polluted days ("H" and "M" samples) and clean days ("L" samples) in different seasons

**10)** Page 10, Lines 21-29: table S1 merits to be added to the manuscript, as not all the numbers are reported in the text.

**Response:** Page 10, Lines 21-29 in the original manuscript (page 11, line 23-31 in the revised manuscript) discuss the seasonal changes of the relative contribution of fossil/non-fossil sources to EC and OC (e.g., $f_{fossil}(EC)$, $f_{bb}(EC)$, $f_{fossil}(OC)$) (So we suppose the reviewer refers to Table S3). We show those data in Table S3 in the original manuscript, which has been moved to the main text as Table 1. The order of Tables in the supplemental material is adapted accordingly.

**11)** Page 11, Line 5: "6.8 ± 6.0 µg m$^{-3}$". Maybe interquartile range is more significant than standard deviation, as the data distribution is not expected to follow a gaussian curve. Same comment for analogous representation of absolute concentrations in the rest of the text (e.g. pag.11, lines 6, 22)

**Response:** In $^{14}$C-based source apportionment of aerosol, it is common to have a small sample size (e.g., N<30), as $^{14}$C measurement is costly and time-consuming. In this study, to get the representative samples for $^{14}$C analysis, a subset of 33 daily samples was pooled into 9 composite samples (See Sect. 2.4.1 for sampling selection for $^{14}$C analysis). The interquartile range (25th–

75th percentile; or Q1–Q3) is not appropriate to measure the spread for those grouped data that are very sparse in the upper range e.g., near Q3, making the estimate vary uncertain (Fig. R3).

[Figure]

Figure R3. An illustration of how to calculate median and interquartile range (Q1–Q3) for $OC_{fossil}$.

We would like to report mean and range of min–max to measure the spread of data. In the revised manuscript, we report $OC_{fossil}$ concentrations:

"OC concentrations from fossil fuel combustion ($OC_{fossil}$) range from about 1 to 20 μg m$^{-3}$, with an average of 6.8 μg m$^{-3}$, which is comparable to non-fossil OC concentrations (range: 2–28 μg m$^{-3}$; mean: 8.2 μg m$^{-3}$)." (page 12, line 7-9)

Analogously, for concentrations in the rest of the text, we now report mean (range of min–max):

"WSOC concentrations from non-fossil sources ($WSOC_{nf}$) are larger than WSOC from fossil sources ($WSOC_{fossil}$) at 95% confidence level (paired $t$-test, $P$-value=0.016), with an average of 5.1 μg m$^{-3}$ (range of 1.5–16.7 μg m$^{-3}$) for $WSOC_{nf}$ versus an average of 3.6 μg m$^{-3}$ (range of 0.6–9.4 μg m$^{-3}$) for $WSOC_{fossil}$ (Fig. 2)." (page 12, line 24-26)

"In winter, the averaged $WIOC_{fossil}$ concentrations of 7.1 μg m$^{-3}$ (range of 3.3–10.1 μg m$^{-3}$) matched the averaged $POC_{fossil}$ concentrations of 6.0 μg m$^{-3}$ (range of 2.7–9.2 μg m$^{-3}$). However, in the warm period, the $WIOC_{fossil}$ concentrations (1.8 μg m$^{-3}$, with a range of 0.8–5.4 μg m$^{-3}$) do not match the estimated $POC_{fossil}$ (2.7 μg m$^{-3}$, with a range of 0.8–7.1 μg m$^{-3}$) equally well." (page 17, line 18-21)

**12)** Pag.11, line 21-23: "larger than", "comparable with": which are the statistical criteria used to evaluate comparability?

**Response:** We perform paired $t$-tests (N=12) to evaluate comparability for $WSOC_{nf}$ vs. $WSOC_{fossil}$, $WIOC_{nf}$ vs. $WIOC_{fossil}$. $WSOC_{nf}$ concentrations are larger than $WSOC_f$ at 95% confidence level (paired $t$-test, $P$-value = 0.016) and the difference between $WIOC_{nf}$ and $WIOC_{fossil}$ is not significant ($P$-value =0.113).

Note that the paired $t$-test might not be strictly valid for WSOC, because there are two data points have a very large difference compare to the rest in the scatter plot of $WSOC_{fossil}$ vs. $WSOC_{nf}$ (Fig.

R4a). But the *P*-value (=0.016) is clearly smaller than 0.05, that we do not expect this to make a big difference in our conclusion.

The revised text shows:

> "WSOC concentrations from non-fossil sources (WSOC$_{nf}$) are larger than WSOC from fossil sources (WSOC$_{fossil}$) at 95% confidence level (paired *t*-test, *P*-value=0.016), with an average of 5.1 µg m$^{-3}$ (range of 1.5–16.7 µg m$^{-3}$) for WSOC$_{nf}$ versus an average of 3.6 µg m$^{-3}$ (range of 0.6–9.4 µg m$^{-3}$) for WSOC$_{fossil}$ (Fig. 2). WIOC concentrations from non-fossil sources (WIOC$_{nf}$) do not differ significantly from fossil sources (WIOC$_{fossil}$) (paired *t*-test, *P*-value=0.113)." (page 12, line 24-28)

[Figure]

**Figure R4. (a)** A scatter plot of WSOC$_{fossil}$ concentrations versus WSOC$_{nf}$ concentrations. **(b)** A scatter plot of WIOC$_{fossil}$ concentrations versus WIOC$_{nf}$ concentrations. Uncertainties of WSOC$_{fossil}$, WSOC$_{nf}$, WIOC$_{fossil}$ and WIOC$_{nf}$ are shown by blue bars.

**13)** Pag.12, line 13: "The fossil OC is less water soluble in winter with lower (WSOC/OC)$_{fossil}$ ratios of around 0.5 than in the warm period". What is "warm period"? Why indicating the value during winter and not during the warm period? Are the differences statistically significant, also considering the limited number of data available?

**Response:** In this study, we use "warm period" to represent spring, summer and autumn, opposite to the cold winter. This is clarified when "warm period" is used for the first time in this manuscript (page 11, line 16). We explain "warm period" again to remind readers in the revised text:

> "The fossil OC is less water soluble in winter with somewhat lower (WSOC/OC)$_{fossil}$ ratios  than in the rest of seasons (i.e., warm period). " (page 13, line 18-19)

The values for (WSOC/OC)$_{fossil}$ ratios (e.g., mean, standard deviation, range) in both winter (0.50 ± 0.03, with a range of 0.48–0.53)  and warm period (0.57 ± 0.08, with a range of 0.42–0.70) have been added to the text.  Considering the limited number of data available (N=3 in winter,

N=9 in warm period) (as the reviewer reminds), a parameter test to compare $(WSOC/OC)_{fossil}$ ratios in winter and warm period  is not applicable. However, ranges and standard deviations allow us to examine the variability of data, and we found that (WSOC/OC)$_{fossil}$ ratios in winter (0.50 ± 0.03, with a range of 0.48–0.53)  fall into the lower end of the range of $(WSOC/OC)_{fossil}$ ratios in warm period (0.57 ± 0.08, with a range of 0.42–0.70). In warm period, the lowest $(WSOC/OC)_{fossil}$ ratio was found for Summer-L (0.42), much lower than that for Summer-H (0.62), which is very likely related to the formation of high pollutant concentrations for Summer-H.  This has been explained in the same paragraph of Sect. 3.2 that more stagnant conditions in polluted periods allow for accumulation of pollutants and also more time for photochemical processing of WIOC and WSOC formation.

The underlined sentences have been added in the revised text (page 13, line 19-20).

**14)** Paragraph 3.3: similarities in $\delta^{13}C$ reference values for different sources affect the results presented here. The results show very high variability and this should be better commented in the text, also in the light of figure S4.

**Response:** In response to question 7, we have moved the Figure S4 to the main text as Figure 6.

We agree with the reviewer that $\delta^{13}C$ reference values for different sources are less well-constrained in contrast to $F^{14}C$ reference values, leading to high variability in EC source apportionment results using Bayesian Markov chain Monte Carlo (MCMC) calculations (for details, see Sect. 2.6).

The results of the MCMC calculations are the posterior probability density functions (PDFs) for the relative contribution from biomass burning ($f_{bb}$), liquid fossil fuel combustion ($f_{liq.fossil}$)  and coal combustion to EC ($f_{coal}$) (Fig. 6). The PDFs of $f_{liq.fossil}$ and $f_{coal}$ are much more spread out than that of $f_{bb}$ , as $f_{bb}$ is well-constrained by $F^{14}C$. The interquartile ranges for $f_{liq.fossil}$ overlap with those for $f_{coal}$ in winter and spring (Table S7). However, comparing the PDFs distribution for both cases give a more complete picture. As shown in Fig. 6, there is fair amount of overlap between the PDFs distributions of $f_{liq.fossil}$ and $f_{ocal}$. Though with some overlaps, in all seasons, the distribution of $f_{liq.fossil}$ are skewed to the left,  while  $f_{coal}$ is skewed to the right, with considerably higher median $f_{liq.fossil}$ than median $f_{coal}$. With the current inherent uncertainties in this state-of-the-art source apportionment methods it will not be possible to draw more firm conclusions than that these probability distributions show a certain trend, despite some possible overlap.

To better separate $f_{liq.fossil}$ from $f_{ocal}$, more measurements of $\delta^{13}C$ of EC from localized emission sources are in urgent need for further studies. In this study, $\delta^{13}C$ source signatures for EC are fully complied and established by a thorough literature search, but unfortunately there are not many studies on $\delta^{13}C$ of EC from different combustion sources and how it changes with combustion conditions.

We have added the above underlined sentences to the Sect. 3.3 (page 14, line 32 to page 15, line 4).

[Figure]

**Figure 6.** Probability density functions (PDFs) of the relative source contributions of **(a)** liquid fossil fuel combustion ($f_{liq.fossil}$), **(b)** coal combustion ($f_{coal}$) and **(c)** biomass burning ($f_{bb}$) to EC constrained by combining radiocarbon and $\delta^{13}C$ measurements, calculated using the Bayesian Markov chain Monte Carlo approach. For details, see Sect. 2.6.

**15)** Pag.13 line 31: "moderately". Quantify and evaluate statistical significance

**Response:** We have quantified the increase in both $EC_{liq.fossil}$ and $EC_{coal}$ from summer to winter and avoid to use "moderately" to evaluate the statistical significance based on the small sample sizes, the revised text shows:

"EC from coal combustion ($EC_{coal}$) has a 5-fold increase from about 0.3 μg m$^{-3}$ in summer and autumn to 1.6 μg m$^{-3}$ in winter. EC from liquid fossil fuel ($EC_{liq.fossil}$) varies less strongly than $EC_{bb}$ and $EC_{coal}$, by 4-times from 0.7 μg m$^{-3}$ in summer and 2.9 μg m$^{-3}$ in winter…Compared to the 4-times increase in $EC_{liq.fossil}$ from summer to winter, $EC_{coal}$ only increases by five times in winter" (page 15 line 11-12)

**16)** Pag.14, line 1: "more constant". Compared to what?

**Response:** We intent to say that winter-time increase in $EC_{coal}$ is only slightly higher than the increase in $EC_{liq.fossil}$. This has been explained in the previous sentence "Compared to the 4-times increase in $EC_{liq.fossil}$ from summer to winter, $EC_{coal}$ only increases by five times in winter". To avoid confusion, we thus delete the sentence:

 (page 15, line 12-13)

**17)** Pag.14, line 24: "rapid". Please quantify (hours? Days?) and justify the sentence.

**Response:** We agree with the reviewer that "rapid" is not enough to quantify how fast is SOC formation. Further, after careful consideration of this statement, we think that it is not justify to conclude "rapid" formation of SOC from "the importance of fossil derived SOC formation to fossil OC during wintertime was also found in other Chinese cities". We thus delete the "suggesting the *rapid* formation of SOC even in winter (R. J. Huang et al., 2014)."

The revised text shows:

"Much higher contribution of $SOC_{fossil}$ to $OC_{fossil}$ (an annual average of around 70%) was found in southern China (Y. L. Zhang et al., 2014). The importance of fossil derived SOC formation to fossil OC during wintertime was also found in other Chinese cities, including Beijing, Shanghai and Guangzhou (Zhang et al., 2015a)." (page 16, line 1-4)

**18)** Pag.15, line 31: "Those contradictions will be discussed in the following section". Coal is hardly mentioned in the following paragraph, thus it is unclear what the authors are referring to.

**Response:** We intend to say "Possible causes of those contradictions will be explained in the following section.", and we have therefore altered the text to specify (page 17, line 11).

In the followed Sect. 3.6, we state that:

(a) "In the warm period, semi-volatile OC from fossil emission sources partitions more readily to the gas-phase leading to lower primary OC/EC ratios compared to winter. This is supported by laboratory studies and ambient observations, which find that the primary OC/EC ratio for vehicle emissions is lower in warm period than in winter (Xie et al., 2017; X. H. H. Huang et al., 2014)." (highlight in yellow on page 18, line 12-15)

That is, primary OC/EC ratios from coal combustion ($r_{coal}$) and vehicle emissions ($r_{vehicle}$) in warm period are lower than that in winter. This can (partially) explain the first contradiction that $(WIOC/EC)_{fossil}$ ratios in warm period indicating vehicle emissions is the overwhelming fossil source, which is inconsistent with the fact both coal combustion and vehicle emissions contribute to fossil EC.

(b) We observed "decreased $(WIOC/EC)_{fossil}$ when pollution gets worse in summer and spring, indicating the loss of fossil WIOC during polluted period. This is probably due to more stagnant conditions in polluted periods, which allows for accumulation of pollutants and also more time for photochemical processing of WIOC and SOC formation" (highlight in yellow on page 17, line 21 to page 18, line 1-3)

This may explain why the differences in $(WIOC/EC)_{fossil}$ between winter and warm period are bigger than $\delta^{13}C_{EC}$ indicated (another contradiction). Because $WIOC_{fossil}$ can be affected by atmospheric processing, while $\delta^{13}C_{EC}$ not.

**19)** Pag.16, line 11 and Pag.17, line 6: "slope of 1.31, and intercept of 0.32 and an $R^2$ of 0.92". "a slope of 0.62, and intercept of 0.01 and an $R^2$ of 0.92". As important uncertainties affect quantities both on x and y axis, 2-sided (Deming) regression should be attempted for better representation of these regression lines

**Response:** The reviewer is right to point out that the regression line is affected by uncertainties both on *x* and *y* axis. Ordinary least squares (OLS) regression (which was used in the original manuscript) is an acceptable approximation for Deming regression if relative uncertainties in *x*-axis are small compared to to relative uncertainties in *y*-axis and/or $R^2$ is high (Wu et al., 2018). Both are the case for our regressions. On the one hand, *x*-axis ($WIOC_{fossil \, or} \, WSOC_{fossil}$; Table S3) has small error relative to *y*-axis ($POC_{fossil}$ or $SOC_{fossil}$; Table S4) as shown in Figs. 9a, 9b. On the other hand, high $R^2$ (>0.9; Figs. 9c, 9d) limits the deviations between Deming regression and OLS regression. As a consequence, Deming regression results are very close to OLS results.

(a) regression of $POC_{fossil}$ (*y*) on $WIOC_{fossil}$ (*x*)

Deming regression results ($y = 1.32x + 0.31$, $R^2 = 0.92$) are very close to OLS results ($y = 1.31 \, x +0.31$, $R^2=0.92$), with very similar slope, intercept and correlation of determination ($R^2$).

(b) regression of $SOC_{fossil}$ (*y*) on $WSOC_{fossil}$ (*x*)

Deming regression results ($y = 0.62x + 0$, $R^2=0.92$) are very close to OLS results ($y = 0.62x + 0.01$, $R^2=0.92$)

In this study, the regression results are not used for further calculation and discussion. If possible, we think it probably acceptable to keep the OLS results as they were in the original manuscript.

**20)** Pag.17, line 10. "that a small fraction of primary fossil OC is water-soluble (Dai et al., 2015; Yan et al., 2017)." This sentence should be moved more above, as it is also a justification of higher fossil POA compared to fossil WIOC.

**Response:** We appreciate this point. We have moved this sentence and the following explanation from the 4th paragraph to the 2nd paragraph of Sect. 3.6.

> "On the other hand, measurements of fresh emissions from fossil sources show that only a small fraction (~10%) of primary fossil OC is water-soluble (Dai et al., 2015; Yan et al., 2017). The differences between $POC_{fossil}$ and $WIOC_{fossil}$ (25–55%) are much larger than that and therefore the small fraction of primary fossil WSOC can not explain the differences between $POC_{fossil}$ and $WIOC_{fossil}$" (page 17, line 27-30)

**21)** Pag.19, line 2: "We suggest that $WIOC_{fossil}$ and $WSOC_{fossil}$ are probably a better approximation for primary and secondary fossil OC, respectively, than $POC_{fossil}$ and $SOC_{fossil}$ estimated using the EC tracer method". This is in contrast with the sentence at the previous point.

**Response:** We thank the reviewer for this helpful comment. Several statements that we made were more ambiguous than intended, and we have clarified the text:

"WIOC$_{fossil}$ and WSOC$_{fossil}$ have been used widely as proxies of the primary and secondary fossil OC, respectively, since primary fossil sources tend to produce mainly WIOC. In winter, mass concentrations of WIOC$_{fossil}$ were comparable to POC$_{fossil}$ and WSOC$_{fossil}$ to SOC$_{fossil}$, where POC$_{fossil}$ and SOC$_{fossil}$ are estimated using EC tracer method. However, the agreement was worse in the warm period, even though the respective concentrations were highly correlated. In other words, variations in WIOC$_{fossil}$ and WSOC$_{fossil}$ follow similar trends as POC$_{fossil}$ and SOC$_{fossil}$, respectively. However, the absolute concentrations of WIOC$_{fossil}$ and WSOC$_{fossil}$ are not equal to those of estimated POC$_{fossil}$ and SOC$_{fossil}$, especially in the warm period." (page 20, line 5-12)

We conclude that "We suggest that WIOC$_{fossil}$ and WSOC$_{fossil}$ are probably a better approximation for primary and secondary fossil OC, respectively, than POC$_{fossil}$ and SOC$_{fossil}$ estimated using the EC tracer method" from the discussion in Sect 3.6 that:

"the most likely explanation for the difference between WIOC$_{fossil}$ and POC$_{fossil}$ is the overestimate of POC$_{fossil}$ by the EC tracer method. POC$_{fossil}$ is calculated by multiplying EC$_{fossil}$ with primary OC/EC ratios for fossil sources ($r_{fossil}$ in Eq. 11). Thus, an overestimate of POC$_{fossil}$ result has two causes. First, $r_{fossil}$ might be overestimated (as EC$_{fossil}$ is well constrained by $^{14}$C), which could result either from a too high estimated fraction of coal burning in the warm period, or through rapid evaporation of POC at warmer temperatures. In the warm period, semi-volatile OC from fossil emission sources partitions more readily to the gas-phase leading to lower primary OC/EC ratios compared to winter. This is supported by laboratory studies and ambient observations, which find that the primary OC/EC ratio for vehicle emissions is lower in warm period than in winter (Xie et al., 2017; X. H. H. Huang et al., 2014). Second, during longer residence time in the atmosphere POC might not be chemically stable and $r_{fossil}$ decreases with aging time in the atmosphere". (page 18, line 7-16)

Further, the large uncertainties in $r_{fossil}$ lead to large uncertainties in the resulted POC$_{fossil}$ and SOC$_{fossil}$, but WIOCfossil and WSOCfossil are well-constraint by $^{14}$C. So we decide to keep this sentence.

**Minor comments:**

**22)** Page 4, line 15: "< 0.2 µg m$^{-2}$) compared to the TC loading of the samples (13–246 µg m$^{-2}$". Replace with "< 0.2 µg **c**m$^{-2}$) compared to the TC loading of the samples (13–246 µg **c**m$^{-2}$"

**Response:** Thank you for spotting out the typo. Corrected (page 4, line 16).

**23)** Pag.5 line 7: "WSOC can be calculated as the difference between OC and WIOC". Unclear why this sentence is here. The previous reference to radiocarbon measurements is confusing (as radiocarbon determination is not carried out as difference, as explained on page 7)

**Response:** Thank you for pointing this out. We have rephrased the sentence to avoid confusion as follows:

"$^{14}$C values of WSOC are calculated from $^{14}$C values of OC and WIOC according to the isotope mass balance (Eq. 4)" (page 5, line 13-14)

**24)** Page 5, line 19: "By water-extraction, water-soluble OC (WSOC) is removed from filter pieces (Dusek et al., 2014)". The role of WSOC removal as a key procedure for reducing the impact of possible pyrolysis on $^{14}$C measurements of EC merits to be better evidenced as a key step for the correct $^{14}$C in EC measurement. In the years 2012-2014 three thermal treatments were developed nearly in parallel and all of them identified WSOC removal as a key step for radiocarbon measurement on EC. Suitable reference should include also Zhang et al, 2012 (https://doi.org/10.5194/acp-12-10841-2012) and Bernardoni et al, 2013 (http://dx.doi.org/10.1016/j.jaerosci.2012.06.001). Please note that these were the methods object of the inter-comparison reported in the mentioned Zenker et al., 2017 papers.

**Response:** We have added Zhang et al. (2012) and Bernardoni et al. (2013) in the text, the revised manuscript shows:

"By water-extraction, water-soluble OC (WSOC) is removed from filter pieces (Zhang et al., 2012; Bernardoni et al., 2013; Dusek et al., 2014)". (page 5, line 27-28)

Accordingly, Zhang et al. (2012) and Bernardoni et al. (2013) have been added to the reference list.

**25)** Page 10, lines 23-24: "Beijing shows a very different seasonal trend, where $f_{bb}$(EC) was lowest in summer (~7%) and increased to ~20% during the rest of the year (Zhang et al., 2017)". Please, introduce the sentence, e.g. "By comparison with literature data for Beijing"

**Response:** Done. The revised text shows:

"By comparison with literature data for Beijing, Beijing shows a very different seasonal trend, where $f_{bb}$(EC) was lowest in summer (~7%) and increased to ~20% during the rest of the year (Zhang et al., 2017)." (page 11, line 26-27)

**26)** Page 10 line 30 (and following): change "around" with "about"

**Response:** Corrected. There are 4 occasions (page 12, line 1, 3, 8; page 15, line 6).

**27)** Pag.16, line 1: "Fossil WIOC (WIOC$_{fossil}$) and WSOC (WSOC$_{fossil}$) has been used". Change into "Fossil WIOC (WIOC$_{fossil}$) and WSOC (WSOC$_{fossil}$) have been used"

**Response:** Corrected (page 17, line 13).

**28)** Pag.16, line 26: "Thus, an overestimate of POC$_{fossil}$ result have two causes". Change into: "Thus, an overestimate of POC$_{fossil}$ result has two causes.

**Response:** Corrected (page 18, line 10).

**29)** Pag. 17, line 27: "An increased contributions". Change into: "An increased contribution"

**Response:** Done (page 19, line 11).


**Response:** Thank you for this comment. To validate this statement, we did the water-extraction for sample Winter-M and Autumn-H to remove WSOC. Then the WIOC amount (referred to as $M_{WIOC\_WE}$) on the water-extracted filter samples was measured directly following the same procedures of non-treated samples.

We find that the directly measured $M_{WIOC\_WE}$ is closer to $M2_{WIOC}$ than to $M1_{WIOC}$ (Table R1). This suggests that it is more likely that WIOC has a similar recovery as OC (i.e., assumption used to estimate $M2_{WIOC}$) rather than 100% recovery (i.e., assumption used to estimate $M1_{WIOC}$).

Due to the limited filter materials, we could not measure $M_{WIOC\_WE}$ for all the samples. In this study, $M_{WIOC}$ is assumed to vary from $M1_{WIOC}$ to $M2_{WIOC}$. The most likely value of $M_{WIOC}$ is chosen at $M1_{WIOC} + 2/3 \times (M2_{WIOC} - M1_{WIOC})$. Once $M_{WIOC}$ is estimated, the $F^{14}C_{(WSOC)}$ can be calculated following the Eq. (4). The best estimate and ranges of $F^{14}C_{(WSOC)}$ is presented in Fig. S2 and Table S1. F14C(WSOC) is only slightly sensitive to MWIOC. If we shift the MWIOC from M1WIOC to M2WIOC, the average values of F14C(WSOC) only change by less than 0.03 (absolute differences). The underlined sentences have been added to the Sect. 2.5 (page 7, line 25–26).

**Table R1.** Mass concentrations of WIOC ($M_{WIOC}$) for sample Winter-M and Autumn-H. See details in estimation of $M1_{WIOC}$ and $M2_{WIOC}$ in Sect. 2.5. In this study, we chose $M1_{WIOC} + 2/3 \times (M2_{WIOC} - M1_{WIOC})$ as the most likely value of $M_{WIOC}$. $M_{WIOC\_WE}$ is the directly measured $M_{WIOC}$ on water-extracted filters.

| Sample Name | $M1_{WIOC}$ ($\mu g\ m^{-3}$) | $M2_{WIOC}$ ($\mu g\ m^{-3}$) | $M1_{WIOC} + 2/3 \times (M2_{WIOC} - M1_{WIOC})$ ($\mu g\ m^{-3}$) | Directly measured $M_{WIOC}$ (referred to as $M_{WIOC\_WE}$) ($\mu g\ m^{-3}$) |
|---|---|---|---|---|
| Winter-M | 13.1 | 18.7 | 16.8 | 18.2 |
| Autumn-H | 9.0 | 11.0 | 10.4 | 10.8 |

6) Page 7, line 14-20. Conversion factors are applied to convert $F^{14}C$ to the relative contribution of non-fossil sources to EC/OC. The conversion factors are $F^{14}C_{bb}$ (= $1.10 \pm 0.05$) for EC and $F^{14}C_{nf}$ (= $1.09 \pm 0.05$) for OC, respectively. Why are the two conversion factors slightly different? I suggest the authors to explain this clearly in the method section.

**Response:** $F^{14}C_{nf}$ ($1.09 \pm 0.05$) for OC is slightly smaller than $F^{14}C_{bb}$ ($1.10 \pm 0.05$) for EC, because except biomass burning, biogenic emissions also contribute to OC, but have a smaller $F^{14}C$ than that of biomass burning. This is clarified in the Method Sect. 2.5:

"F14Cbb represents F14C of biomass burning including wood burning and crop residue burning." (page 7, line 29)

"$F^{14}C_{nf}$ is $F^{14}C$ of non-fossil sources include both biomass burning and biogenic emissions." (page 8, line 9-10)

"$F^{14}C_{nf}$ is thus estimated as 1 .09 ± 0.05 (Lewis et al., 2004; Levin et al., 2010; Y. L. Zhang et al., 2014). The central value of 1.09 corresponds to 15% contribution of biogenic OC to OC." (page 8, line 13-14)

**7)** Page 9, line 15-16. Are the measurement uncertainties of $F^{14}C_{(EC)}$ and $\delta\ ^{13}C_{EC}$ considered in the MCMC calculations?

**Response:** The measurement uncertainties of $F^{14}C_{(EC)}$ and $\delta\ ^{13}C_{EC}$ are inputs of MCMC and thus are considered in the MCMC calculations. This is clarified in the Method Sect. 2.6 by adding the following underlined sentence:

"The MCMC technique takes into account the variability in the source signatures of $F^{14}C$ and $\delta^{13}C$ (Parnell et al., 2010, 2013), where $\delta^{13}C$ introduces a larger uncertainty than $F^{14}C$. Uncertainties of $\delta^{13}C_{bb}$, $\delta^{13}C_{liq.fossil}$, $\delta^{13}C_{coal}$ and $F^{14}C_{bb}$ as well as the measured ambient $\delta^{13}C_{EC}$ and $F^{14}C_{(EC)}$ are propagated." (page 10, line 15-16)

Consequently, we add the uncertainties of $\delta^{13}C_{EC}$ in Table S1.

**Technical comments:**

**8)** Page 4, line 1. "a" between "in" and "pre-baked" should be deleted.

**Response:** Corrected. (page 4, line 2)

**9)** Page 8, line 8. To be consistent with the text, I think it should a comma in "$OC_{o,nf}$" in Eq. (8). Please check all instances

**Response:** Thank you for your careful reading. This is corrected in Eq. (8).

There are also several occasions in the rest of the manuscript, and the revised manuscript shows:

"As for OC from secondary origin (i.e., $SOC_{fossil}$ and $OC_{o,nf}$)," (page 16, line 5)

"**Figure 7.** **(a)** The estimated mass concentrations of $POC_{bb}$, $OC_{o,nf}$, $POC_{fossil}$, $SOC_{fossil}$ (µg m$^{-3}$) in total OC of PM$_{2.5}$ samples. The error bars indicate the interquartile range (25th–75th percentile) of the median values. **(b)** The percentage of $POC_{bb}$, $OC_{o,nf}$, $POC_{fossil}$, $SOC_{fossil}$ in total OC. **(c)** Average source apportionment results of OC in each season and over the year. The numbers below the pie charts represent the seasonally/annually averaged OC concentrations." (page 35)

"**Figure S3.** **(a)** An example probability density functions **(PDFs)** of concentrations of $POC_{fossil}$ (red), $SOC_{fossil}$ (light blue) for sample Autumn-L. **(b)** PDFs of concentrations of and $OC_{o,nf}$ (light blue) and $POC_{bb}$ (red) for the same sample." (page S6 in the Supplement)

R21

**10).** Page 8, line 11. A citation is missing for the statement that "In most cases, contributions of primary biogenic OC to PM$_{2.5}$ are likely small".

**Response:** We add Gelencsér et al. (2007) and Guo et al. (2012). (page 9, line 6)

The new citations are included in the revised reference list:

Gelencsér, A., May, B., Simpson, D., Sánchez-Ochoa, A., Kasper-Giebl, A., Puxbaum, H., Caseiro, A., Pio, C., and Legrand, M.: Source apportionment of PM$_{2.5}$ organic aerosol over Europe: Primary/secondary, natural/anthropogenic, and fossil/biogenic origin, J. Geophys. Res., 112, D23S04, doi:10.1029/2006JD008094, 2007.

Guo, S., Hu, M., Guo, Q., Zhang, X., Zheng, M., Zheng, J., Chang, C. C., Schauer, J. J., and Zhang, R.: Primary sources and secondary formation of organic aerosols in Beijing, China, Environ. Sci. Technol., 46, 9846–9853, 2012.

**11)** Page 8, line 19. It should be "combining" instead of "combing".

**Response:** Thank you for spotting this typo. Corrected (page 9, line 14).

**12)** Page 8, line 25. Give full name of PDF, because it is used for the first time in this manuscript. The authors should check the manuscript again for proper use of abbreviations.

**Response:** We now explain PDF the first time it is used in this revised manuscript:

"For $p$ values, random values from the respective probability density function (PDF) of $p$ were used" (page 9, line 20-21)

After the first appearance of the abbreviation, the abbreviation PDF is used in the rest of the manuscript:

"The results of the MCMC calculations are the posterior PDFs for $f_{bb}$, $f_{liq.fossil}$ and $f_{coal}$" (page 10, line 16-17)

**13)** Page 10, line 22. "a slightly lower value" instead of "a slightly lower values"

**Response:** In response of question (9) from reviewer #1, to be specify, we revised the text and delete the sentence "with a slightly lower value in spring (14 ± 3%)":

"$f_{bb}$(EC) is higher in winter (28 ± 4%) than that in other seasons (i.e., warm period, on average 15 ± 2 %). " (page 11, line 24-25)

**14)** Page 17, line 23. "various carbonaceous aerosol fractions"

**Response:** Done (page 19, line 7).

**15)** Page 17, line 27. "An increased contribution of non-fossil sources to all carbon fractions was observed"

**Response:** Corrected (page 19, line 11).

---

## Author Response (AR2)

**Comments to the Author:**

The authors have reasonably well addressed the comments of the two anonymous referees and they have modified their manuscript accordingly. However, the comments given below should be addressed and several alterations are needed for the Main text and Supplement before the manuscript can be published in ACP.

**Response:** Thank you for providing us the opportunity to revise and improve our manuscript. The comments on the main text and supplement are addressed accordingly. Detailed responses to the comments are provided in blue. Attached please also find the marked-up manuscript to track the changes in the revised manuscript.

**1. Comments on the main text and supplement:**

**Main text:**

Page 8, line 23, and also further within the text: the "EC tracer method" is mentioned without any literature reference. At least one literature reference is needed for it. I suggest including the following literature reference here: "Turpin and Huntzicker, 1995"; the full reference for it is: Turpin, B. J. and Huntzicker, J. J.: Identification of secondary organic aerosol episodes and quantitation of primary and secondary organic aerosol concentrations during SCAQS, Atmos. Environ., 29, 3527-3544, 1995.

**Response:** The citation is added in Method Sect. 2.5 (page 8, line 23-24) and in the Reference list (page 25, line 29-30).

Page 9, line 19: "0 at the lower limit and upper limit" is unclear to me. How can it be 0 at the upper limit?

**Response:** Random values for $F^{14}C_{bb}$, $F^{14}C_{nf}$, $r_{bb}$, $r_{coal}$ and $r_{vehicle}$ are estimated as follows: for each parameter we estimated a central value and a lower and upper limit as described in Sect. 2.5. For example, central value for $r_{bb}$ is estimated as 4, with lower limit of 3 and higher limit of 5. The random values are then chosen from a triangular frequency distribution, which has its maximum frequency at the central value of $r_{bb}$, and 0 frequencies at the upper and lower limits of $r_{bb}$, as shown in Fig. R1.

To make this clear, we revised the text to (changes are underlined):

"For $F^{14}C_{bb}$, $F^{14}C_{nf}$, $r_{bb}$, $r_{coal}$ and $r_{vehicle}$ random values of each parameter are chosen from a triangular frequency distribution, which has its maximum frequency at the central value and 0 frequency at the lower limit and upper limit of each parameter." (page 9, line 18-20)

[Figure]

Figure R1. A triangular frequency distribution (n=10000) for $r_{bb}$.

Page 24, line 42: The reference is incomplete.

**Response:** The article number and doi number are added for the " Sheesley et al., 2012":

"Sheesley, R. J., Kirillova, E., Andersson, A., Kruså, M., Praveen, P., Budhavant, K., Safai, P. D., Rao, P., and Gustafsson, Ö.: Year-round radiocarbon-based source apportionment of carbonaceous aerosols at two background sites in South Asia, J. Geophys. Res.-Atmos., 117, D10202, doi:10.1029/2011JD017161, 2012. "

Page 25, line 42: The reference is incomplete.

**Response:** We add the article number and doi number:

"Winiger, P., Andersson, A., Eckhardt, S., Stohl, A., and Gustafsson, Ö.: The sources of atmospheric black carbon at a European gateway to the Arctic, Nat. Commun., 7, 12776, https://doi.org/10.1038/ncomms12776, 2016. "

Page 26, line 9: The reference is incomplete.

**Response:** We add the article number and doi number:

"Yan, C., Zheng, M., Bosch, C., Andersson, A., Desyaterik, Y., Sullivan, A. P., Collett, J. L., Zhao, B., Wang, S., He, K. and Gustafsson, Ö.: Important fossil source contribution to brown carbon in Beijing during winter, Sci. Rep., 7, 43182, https://doi.org/10.1038/srep43182, 2017. "

Line 56: Replace "of and" by "of".

**Response:** Done.

We found that the legend title of Figure S3 is incorrect. We thus corrected the legend in the revised Supplement. (page S7)

**2. Response:** Alterations and corrections are made following the co-editor's comments. Below are the alterations made in the main text and supplement.

**For the Main text:**

Page 2, line 12: Replace "aaccording" by "according".
Page 3, line 16: Delete "one-year" to be consistent with the reply to Main comment 1 of Anonymous reviewer #2.
Page 5, line 5: Replace "EC step" by "the EC step".
Page 5, line 11: Replace "total carbon (TC = OC + EC)" by "TC" as TC was already defined in line 13 of page 4.
Page 5, line 22: Replace "with dry" by "with a dry".
Page 5, line 28: Replace "that left" by "that is left".
Page 6, line 2: Replace "by EUSAAR_2" by "by the EUSAAR_2".
Page 7, line 6: Insert a space before "In 2010".
Page 7, line 18: Replace "with the" by "by the".
Page 7, line 25: Replace "is presented" by "are presented".
Page 7, line 27: Replace "dividing with" by "dividing by".
Page 8, line 1: Replace "because that" by "because".
Page 8, line 10: Replace "include both" by "including both".
Page 8, line 11: Replace "Levin and Hammer, 2013; Levin et al., 2010" by "Levin et al., 2010, 2013".
Page 8, line 13: Replace "1 .09" by "1.09".
Page 8, line 26: Replace "ealier" by "earlier".
Page 9, line 19: Replace "and is 0" by "and 0".
Page 9, line 23: Replace "estimates and the interquartile range" by "estimate and the interquartile ranges".
Page 11, line 15: Replace "also to be slightly" by "also slightly".
Page 11, line 22: Replace "than that in" by "than in".
Page 12, line 2: Replace "is larger" by "are larger".
Page 12, line 12: Replace "that to" by "those to".
Page 12, line 23: Insert a space before "(range of".
Page 13, line 7: Replace "as at an" by "as an".
Page 13, line 29: Replace "type of" by "types of".
Page 14, lines 20-21: Replace "Markov chain Monte Carlo techniques (MCMC)" by "MCMC" as MCMC was already defined in line 27 of page 9.
Page 15, line 9: Replace "than that in" by "than in".
Page 15, line 11: Replace "in pervious" by "in previous".
Page 15, line 28: Replace "than winter" by "than in winter".
Page 16, line 24: Replace "period is also" by "period are also".
Page 16, line 26: Replace "ratios was found" by "ratios were found".
Page 16, line 31: Replace "that is more" by "that are more".
Page 17, line 1: Replace "In warm" by "In the warm".
Page 17, line 10: Replace "are mainly" by "is mainly".

Page 17, line 13: Replace "wide range" by "the wide range".
Page 17, line 20: Replace "most of samples" by "most samples".
Page 17, line 21: Replace "of WIOC" by "of the WIOC".
Page 18, line 5: Replace "overestimate of" by "overestimate of the".
Page 18, line 9: Replace "in warm" by "in the warm".
Page 18, line 19: Replace "in warm" by "in the warm".
Page 19, line 3: Replace "were observed" by "was observed".
Page 19, line 6: Replace "Markov chain Monte Carlo techniques (MCMC)" by "MCMC" as MCMC was already defined in line 27 of page 9.
Page 19, line 17: Replace "than that for" by "than for".
Page 19, line 19: Replace "was found" by "were found".
Page 19, line 23: Replace "in warm" by "in the warm".
Page 19, line 31: Replace "using EC" by "using the EC".
Page 20, line 3: Replace "resulted" by "resulting".
Page 20, line 5: Replace "than other" by "than in the other".
Page 20, line 8: Replace "that photochemical" by "photochemical" and replace "affect final" by "affects final".
Page 20, line 12: Replace "Supplment" by "Supplement".
Page 20, line 16: Replace "preformed" by "performed".
Page 23, line 8: Replace "B. Kromer" by "Kromer, B.".
Page 23, lines 8-14: "Levin et al., 2013" should come after "Levin et al., 2010".
Page 24, line 3: Replace "2006" by "2006.".
Page 25, line 22: Replace "2009." by "2009,".
Page 31, line 2: Replace "4. The" by "4.".
Page 34, line 2: Replace "The estimated" by "Estimated".
Page 35, line 2: Replace "A scatter" by "Scatter".
Page 35, line 3: Replace "and warm" by "and the warm" and replace "The WIOC" by "WIOC".
Page 36, line 3: Replace "with panel" by "as panel" and replace "A scatter" by "Scatter".
Page 36, line 4: Replace "A scatter" by "Scatter".

**For the Supplement:**
Line 20: Replace "of EC" by "of the EC".
Line 55: Replace "An example" by "Example".

[revised manuscript text omitted]

In this study, we used a two-step method (OC step: 375 °C for 3 h; EC step: 850 °C for 5 h) to isolate OC and EC for $\delta^{13}$C analysis, as described in Sect. 2.3. Our earlier study in Xi'an found that EC recovery for $\delta^{13}$C analysis (relative to EC quantified by the thermal-optical reflectance protocol IMPROVE_A; Chow et al., 2007) was on average $123 \pm 8$ %, higher than 100% (Zhao et al., 2018). The reason is that pyrolyzed OC (formed through charring during the OC removal procedure) and possibly some remaining OC compounds (e.g., high molecular weight refractory carbon) can be released at the high temperature of the EC step.

The resulted $\delta^{13}$C of EC could be biased by $\delta^{13}$C of pyrolyzed OC, if the contribution from pyrolyzed OC to the isolated EC is high and $\delta^{13}$C of pyrolyzed OC is very different from $\delta^{13}$C of pure EC. To examine the effect of pyrolyzed OC on $\delta^{13}$C of EC, a sensitivity analysis is performed. $\delta^{13}$C of pyrolyzed OC is not known, but our recent studies suggest that $\delta^{13}$C of pyrolyzed OC is not very different from $\delta^{13}C_{OC}$ (<1‰ in many cases). We thus use $\delta^{13}C_{OC}$ to represent $\delta^{13}$C of pyrolyzed OC. $\delta^{13}$C of pure EC is calculated based on isotope mass balance. This analysis shows that 
[revised manuscript text omitted]